# Descending through a Crowded Valley — Benchmarking Deep Learning Optimizers

## Abstract

Choosing the optimizer is considered to be among the most crucial design decisions in deep learning, and it is not an easy one. The growing literature now lists hundreds of optimization methods. In the absence of clear theoretical guidance and conclusive empirical evidence, the decision is often made based on anecdotes. In this work, we aim to replace these anecdotes, if not with a conclusive ranking, then at least with evidence-backed heuristics. To do so, we perform an extensive, standardized benchmark of more than a dozen particularly popular deep learning optimizers while giving a concise overview of the wide range of possible choices. Analyzing almost 35,000 individual runs, we contribute the following three points: (i) Optimizer performance varies greatly across tasks. (ii) We observe that evaluating multiple optimizers with default parameters works approximately as well as tuning the hyperparameters of a single, fixed optimizer. (iii) While we can not discern an optimization method clearly dominating across all tested tasks, we identify a significantly reduced subset of specific algorithms and parameter choices that generally lead to competitive results in our experiments. This subset includes popular favorites and some lesser-known contenders. We have open-sourced all our experimental results, making them directly available as challenging and well-tuned baselines.[1] This allows for more meaningful comparisons when evaluating novel optimization methods without requiring any further computational efforts.

## 1 Introduction

Large-scale stochastic optimization drives a wide variety of machine learning tasks. Because choosing the right optimization algorithm and effectively tuning its hyperparameters heavily influences the training speed and final performance of the learned model, doing so is an important, every-day challenge to practitioners. Hence, stochastic optimization methods have been a focal point of research (cf. Figure 1), engendering an ever-growing list of algorithms, many of them specifically targeted towards deep learning. The hypothetical machine learning practitioner who is able to keep up with the literature now has the choice among hundreds of methods (cf. Table 2 in the appendix)—each with their own set of tunable hyperparameters—when deciding how to train their model.

There is limited theoretical analysis that would clearly favor one of these choices over the others. Some authors have offered empirical comparisons on comparably small sets of popular methods (e.g. Wilson et al., 2017; Choi et al., 2019; Sivaprasad et al., 2020); but for most algorithms, the only formal empirical evaluation is offered by the original work introducing the method. Many practitioners and researchers, meanwhile, rely on personal and anecdotal experience, and informal discussion on social media or with colleagues. The result is an often unclear, perennially changing "state of the art" occasionally driven by hype. The key obstacle for an objective benchmark is the combinatorial cost of such an endeavor posed by comparing a large number of methods on a large number of problems, with the high resource and time cost of tuning each method's parameters and repeating each (stochastic) experiment repeatedly for fidelity.

Offering our best attempt to construct such a comparison, we conduct a large-scale benchmark of optimizers to further the debate about deep learning optimizers, and to help understand how the choice of optimization method and hyperparameters influences the training performance. Specifically,

---

[1] https://github.com/AnonSubmitter3/Submission543

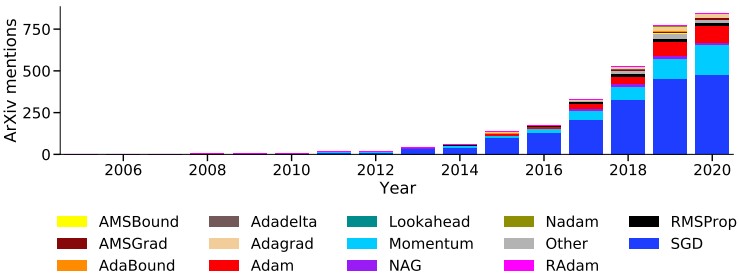

Figure 1: Number of times ArXiv titles and abstracts mention each optimizer per year. All non-selected optimizers from Table 2 in the appendix are grouped into *Other*.

we examine whether recently proposed methods show an improved performance compared to more established methods such as SGD or ADAM. Additionally, we are interested in assessing whether optimization methods with well-working default hyperparameters exist that are able to keep up with tuned optimization methods. To this end, we evaluate more than a dozen optimization algorithms, largely selected for their perceived popularity, on a range of representative deep learning problems (see Figure 4) drawing conclusions from tens of thousands of individual training runs.

Right up front, we want to state clearly that it is impossible to include all optimizers (cf. Table 2 in the appendix), and to satisfy any and all expectations readers may have on tuning and initialization procedures, or the choice of benchmark problems—not least because everyone has different expectations in this regard. In our *personal opinion*, what is needed is an empirical comparison by a third party not involved in the original works. As a model reader of our work, we assume a careful practitioner who does not have access to near-limitless resources, nor to a broad range of personal experiences. As such, the core contributions (in order of appearance, not importance) of our work are:

**A concise summary of optimization algorithms and schedules**    A partly automated, mostly manual literature review provides a compact but extensive list of recent advances in stochastic optimization. We identify more than a hundred optimization algorithms (cf. Table 2 in the appendix) and more than 20 families of hyperparameter schedules (cf. Table 3 in the appendix) published at least as pre-prints.

**An extensive optimizer benchmark on deep learning tasks**    We conduct a large-scale optimizer benchmark, specifically focusing on optimization problems arising in deep learning. We evaluate 14 optimizers on eight deep learning problems using four different schedules, tuning over dozens of hyperparameter settings, to our knowledge, this is the most comprehensive empirical evaluation of deep learning optimizers to date (cf. Section 1.1 on related work).

**An analysis of thousands of optimization runs**    Our empirical experiments indicate that an optimizer's performance highly depends on the test problem (see Figure 4). But some high-level trends emerge, too: (1) Evaluating multiple optimizers with default hyperparameters works approximately as well as tuning the hyperparameters for a fixed optimizer. (2) Using an additional untuned learning rate schedule helps on average, but its effect varies greatly depending on the optimizer and the test problem. (3) While there is no optimizer that clearly dominates across all tested workloads, some of the algorithms we tested exhibited highly variable performance. Others demonstrated decent performance consistently. We deliberately refrain from recommending a single one among them, because we could not find a clear winner with statistical confidence.

**An open-source baseline for future optimizer benchmarks**    Our results are accessible online in an open and easily accessible form (see footnote on Page 1). These results can thus be used as competitive and well-tuned baselines for future benchmarks of new algorithms, drastically reducing the amount of computational budget required for a meaningful optimizer comparison. Our baselines can easily be expanded, and we encourage others to contribute to this collection.

The high-level result of our benchmark is, perhaps expectedly, *not* a clear winner. Instead, our comparison shows that, while some optimizers are frequently decent, they also generally perform similarly, switching their relative positions in the ranking which can partially be explained by the

No Free Lunch Theorem (Wolpert & Macready, 1997). A key insight of our comparison is that a practitioner with a new deep learning task can expect to do about *equally well* by taking almost any method from our benchmark and *tuning* it, as they would by investing the same computational resources into running a set of optimizers with their *default* settings and picking the winner.

Possibly the most important takeaway from our comparison is that "there are now enough optimizers." Methods research in stochastic optimization should focus on *significant* (conceptual, functional, performance) improvements—such as methods specifically suited for certain problem types, inner-loop parameter tuning or structurally novel methods. We make this claim not to discourage research but, quite on the contrary, to offer a motivation for more meaningful, non-incremental research.

## 1.1 RELATED WORK

Following the rapid increase in publications on optimizers, *benchmarking* these methods for the application in deep learning has only recently attracted significant interest. Schneider et al. (2019) introduced a benchmarking framework called DEEPOBS, which includes a wide range of realistic deep learning test problems together with standardized procedures for evaluating optimizers. Metz et al. (2020) presented TASKSET, another collection of optimization problems focusing on smaller but many more test problems. For the empirical analysis presented here, we use DEEPOBS as it provides optimization problems closer to real-world deep learning tasks. In contrast to our evaluation of *existing* methods, TASKSET and its analysis focuses on meta-learning *new* algorithms or hyperparameters.

Both Choi et al. (2019) and Sivaprasad et al. (2020) analyzed specific aspects of benchmarking process. Sivaprasad et al. (2020) used DEEPOBS to illustrate that the relative performance of an optimizer depends significantly on the used hyperparameter tuning budget. The analysis by Choi et al. (2019) supports this point, stating that "the hyperparameter search space may be the single most important factor explaining the rankings." They further stress a hierarchy among optimizers, demonstrating that, given sufficient hyperparameter tuning, more general optimizers can never be outperformed by special cases. In their study, however, they manually chose a hyperparameter search space *per optimizer and test problem* basing it either on prior published results, prior experiences, or pre-tuning trials. Here we instead aim to identify well-performing optimizers in the case of a less extensive tuning budget and especially when there is no prior knowledge about well-working hyperparameter values for each specific test problem. We further elaborate on the influence of our chosen hyperparameter search strategy in Section 4 discussing the limitations of our empirical study.

Our work is also related to empirical generalization studies of adaptive methods, such as that of Wilson et al. (2017) which sparked an extensive discussion whether adaptive methods (e.g. ADAM) tend to generalize worse than standard first-order methods (i.e. SGD).

## 2 BENCHMARKING PROCESS

Any benchmarking effort requires tricky decisions on the experimental setup that influence the result. Evaluating on a specific task or picking a certain tuning budget, for example, may favor or disadvantage certain algorithms (Sivaprasad et al., 2020). It is impossible to avoid these decisions or to cover all possible choices. Aiming for generality, we evaluate the performance on eight diverse real-world deep learning problems from different disciplines (Section 2.1). From a collection of more than a hundred deep learning optimizers (Table 2 in the appendix) we select 14 of the most popular and most promising choices (cf. Figure 1) for this benchmark (Section 2.2). For each test problem and optimizer we evaluate all possible combinations of three different tuning budgets (Section 2.3) and four selected learning rate schedules (Section 2.4), thus covering the following combinatorial space:

$$
\begin{array}{cccc}
\textbf{Problem} & \textbf{Optimizer} & \textbf{Tuning} & \textbf{Schedule} \\
\left\{\begin{array}{c} P1 \\ P2 \\ \dots \\ P8 \end{array}\right\}_{8} \times & \left\{\begin{array}{c} \text{AMSBound} \\ \text{AMSGrad} \\ \dots \\ \text{SGD} \end{array}\right\}_{14} \times & \left\{\begin{array}{c} \text{one-shot} \\ \text{small budget} \\ \text{large budget} \end{array}\right\}_{3} \times & \left\{\begin{array}{c} \text{constant} \\ \text{cosine decay} \\ \text{cosine warm restarts} \\ \text{trapezoidal} \end{array}\right\}_{4} .
\end{array}
$$

Combining those options results in 1,344 possible configurations and roughly 35,000 individual runs.

Table 1: Summary of test problems used in our experiments. The exact model configurations can be found in the work of Schneider et al. (2019).

|     | Data set | Model | Task | Metric | Batch size | Budget *in epochs* |
|-----|----------|-------|------|--------|-----------|-------------------|
| **P1** | Artificial data set | Noisy quadratic | Minimization | Loss | 128 | 100 |
| **P2** | MNIST | VAE | Generative | Loss | 64 | 50 |
| **P3** | Fashion-MNIST | Simple CNN: *2c2d* | Classification | Accuracy | 128 | 100 |
| **P4** | CIFAR-10 | Simple CNN: *3c3d* | Classification | Accuracy | 128 | 100 |
| **P5** | Fashion-MNIST | VAE | Generative | Loss | 64 | 100 |
| **P6** | CIFAR-100 | *All-CNN-C* | Classification | Accuracy | 256 | 350 |
| **P7** | SVHN | *Wide ResNet 16-4* | Classification | Accuracy | 128 | 160 |
| **P8** | War and Peace | RNN | NLP | Accuracy | 50 | 200 |

## 2.1 TEST PROBLEMS

We consider the eight optimization tasks summarized in Table 1, available as the "small" (P1–P4) and "large" (P5–P8) problem sets, respectively, together forming the default collection of DEEPOBS. A detailed description of these problems, including architectures, training parameters, etc. can be found in the work of Schneider et al. (2019).[2] DEEPOBS' test problems provide several performance metrics, including the training and test loss, the validation accuracy, etc. While these are all relevant, any comparative evaluation of optimizers requires picking only a few, if not just one particular performance metric. For our analysis (Section 3), we focus on the final test accuracy (or the final test loss, if no accuracy is defined for this problem). This metric captures, for example, the optimizer's ability to generalize and is thus highly relevant for practical use. Our publicly released results include all metrics for completeness. An example of training loss performance is shown in Figure 16 in the appendix. Accordingly, the tuning (Section 2.3) is done with respect to the validation metric. We discuss possible limitations resulting from these choices in Section 4.

## 2.2 OPTIMIZER SELECTION

In Table 2 in the appendix we collect over a hundred optimizers introduced for, suggested for, or used in deep learning. This list was manually and incrementally collected by multiple researchers trying to keep up with the field over recent years. It is thus necessarily incomplete, although it may well represent one of the most exhaustive of such collections. Even this incomplete list, though, contains too many entries for a meaningful benchmark with the degrees of freedom collected above. This is a serious problem for research: Even an author of a new optimizer, let alone a practitioner, could not possibly be expected to compare their work with every possible competing method.

We thus selected a subset of 14 optimizers, which we consider to be currently the most popular choices in the community (see Table 4 in the appendix). These do not necessarily reflect the "best" algorithms, but are either commonly used by practitioners and researchers, or have recently generated enough attention to garner interest. Our selection is focused on first-order optimization methods, both due to their prevalence for non-convex continuous optimization problems in deep learning as well as to simplify the comparison. Whether there is a significant difference between these optimizers or if they are inherently redundant is one of the questions this work investigates.

With our list, we tried to focus on optimization algorithms over techniques, although we acknowledge, the line being very blurry. Techniques such as averaging weights (Izmailov et al., 2018, e.g.) or ensemble methods (Garipov et al., 2018, e.g.) have been shown to be simple but effective at improving the optimization performance. Those methods, however, can be applied to all methods in our lists, similar to regularization techniques, learning rate schedules, or tuning methods and we have, therefore, decided to omit them from Table 2.

---

[2]All experiments were performed using version `1.2.0-beta` of DEEPOBS and TensorFlow version `1.15` Abadi et al. (2015).

## 2.3 TUNING

**Budget** Optimization methods for deep learning regularly expose hyperparameters to the user. The user sets them either by relying on the default suggestion; using experience from previous experiments; or using additional tuning runs to find the best-performing setting. All optimizers in our benchmark have tunable hyperparameters, and we consider three different *tuning budgets*.

The first budget consists of just a single run. This *one-shot* budget uses the default values proposed by the original authors, where available (Table 4 in the appendix lists the default parameters). If an optimizer performs well in this setting, this has great practical value, as it drastically reduces the computational resources required for training. The other budgets consist of 25 and 50 tuning runs for what we call the *small* and *large budget* settings, respectively.

We only use a single seed for tuning, then repeat the best setting 10 times using different seeds. This allows us to report standard deviations in addition to means, assessing stability. Progressing in this way has the "feature" that our tuning process can sometimes pick "lucky" seeds, which do not perform as well when averaging over multiple runs. This is arguably a good reflection of reality. Stable optimizers should be preferred in practice, which is thus reflected in our benchmark. See Appendix C for further analysis. By contrast, using all 10 random seeds for tuning as well would drastically increase cost, not only for this benchmark, rendering it practically infeasible, but also as an approach for the practical user. Appendix D explores this aspect further: If anything, re-tuning would further *broaden* the distribution of results.

**Tuning method** We tune parameters by random search, for both the small and the large budget. Random search is a common choice in practice due to its efficiency advantage over grid search (Bergstra & Bengio, 2012) and its ease of implementation and parallelization compared to Bayesian optimization (see also Section 4). A minor complication of random search is that the sampling distribution affects the optimizer's performance. One can think of the sampling distribution as a prior over good parameter settings, and bad priors consequently ruin performance. We followed the mathematical bounds and intuition provided by the optimizers' authors for relevant hyperparameters. The resulting sampling distributions can be found in Table 4 in the appendix. In case there is no prior knowledge provided in the cited work we chose similar distributions for similar hyperparameters across different optimizers. Even though a hyperparameter might have a similar naming throughout different optimization algorithms (e.g. learning rate $\alpha$), its appropriate search space can differ across optimizers. Without grounded heuristics on how the hyperparameters differ between optimizers, the most straightforward approach for any user is to use the same search space.

**What should be considered a hyperparameter?** There's a fuzzy boundary between (tunable) hyperparameters and (fixed) design parameters. A recently contentious example is the $\varepsilon$ in adaptive learning rate methods like ADAM. It was originally introduced as a safeguard against division by zero, but has recently been re-interpreted as a problem-dependent hyperparameter choice (see Choi et al. (2019) for a discussion). Under this view, one can actually consider several separate optimizers called ADAM: From an easy-to-tune but potentially limited ADAM$_\alpha$, only tuning the learning rate, to the tricky-to-tune but all-powerful ADAM$_{\alpha,\beta_1,\beta_2,\varepsilon}$, which subsumes SGD as a corner case in its hyperparameter space. In our benchmark, we include ADAM$_{\alpha,\beta_1,\beta_2}$ as a popular choice. While they share the same update rule, we consider them to be different optimizers.

## 2.4 SCHEDULES

The literature on learning rate schedules is now nearly as extensive as that on optimizers (cf. Table 3 in the appendix). *In theory*, schedules can be applied to all hyperparameters of an optimization algorithm but to keep our configuration space feasible, we only apply schedules to the learning rate, by far the most popular practical choice (Goodfellow et al., 2016; Zhang et al., 2020). We choose four different learning rate schedules, trying to cover all major types of schedules (see Appendix E):

- A *constant* learning rate schedule;
- A *cosine decay* (Loshchilov & Hutter, 2017) as an example of a smooth decay;
- A *cosine with warm restarts* schedule (Loshchilov & Hutter, 2017) as a cyclical schedule;
- A *trapezoidal* schedule (Xing et al., 2018) from the warm-up schedules (Goyal et al., 2017).

## 3 RESULTS

**How well do optimizers work out-of-the-box?**    By comparing each optimizer's one-shot results against the tuned versions of all 14 optimizers, we can construct a $14 \times 14$ matrix of performance gains. Figure 2 illustrates this on five test problems showing improvements by a positive sign and a green cell. Detailed plots for all problems are in Figures 9 and 10 in the appendix. For example, the bottom left cell of the largest matrix in Figure 2 shows that AMSBOUND *(1)* tuned using a small budget performs $2.5\%$ better than SGD *(14)* with default parameters on this specific problem.

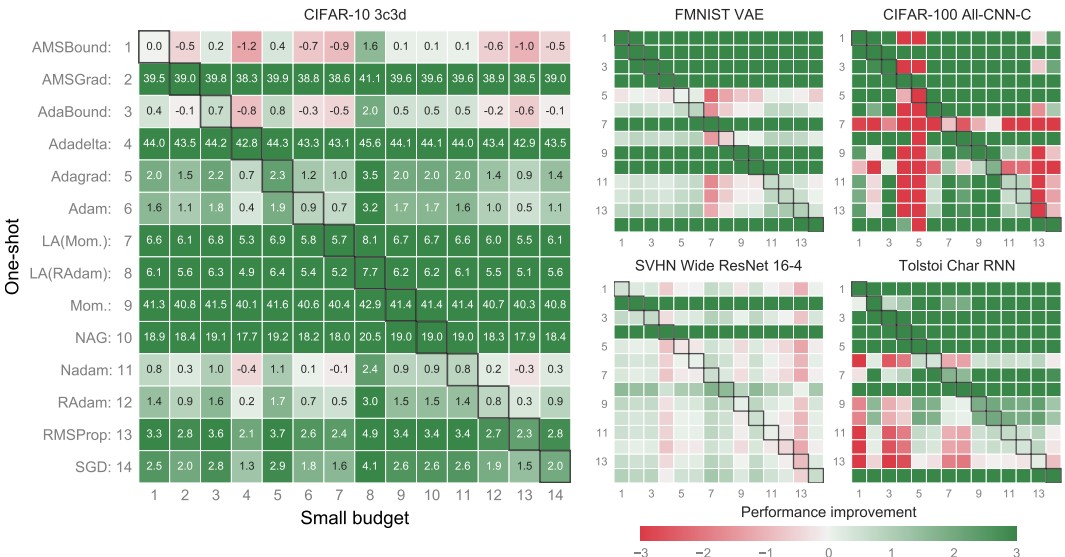

Figure 2: The absolute test set performance improvement after switching from any untuned optimizer (*y*-axis, *one-shot*) to any tuned optimizer (*x*-axis, *small budget*) as an average over 10 random seeds for the *constant* schedule. We discuss the unintuitive occurrence of negative diagonal entries in Appendix F. The colormap is capped at $\pm 3$ to improve presentation, although larger values occur.

A **green row** in Figure 2 indicates that an optimizer's default setting is performing badly, since it can be beaten by any well-tuned competitor. We can observe badly-performing default settings for MOMENTUM, NAG and SGD, advocating the intuition that non-adaptive optimization methods require more tuning, but also for AMSGRAD and ADADELTA. This is just a statement about the default parameters suggested by the authors or the popular frameworks, well-working default parameters might well exist for those methods. Conversely, a **white & red row** signals a well-performing default setting, since even tuned optimizers cannot significantly outperform this algorithm. ADAM, NADAM and RADAM, as well as AMSBOUND and ADABOUND all have white or red rows on several (but not all!) test problems, supporting the rule of thumb that adaptive methods have well-working default parameters. Conversely, **green** (or **red**) **columns** highlight optimizers that, when tuned, perform better (or worse) than all untuned optimization methods. We do not observe such columns consistently across tasks. This supports the conclusion that an optimizer's performance is heavily problem-dependent and that there is no single *best* optimizer across workloads.

Figures 9 to 12 in the appendix and our conclusions from them suggest an interesting alternative approach for machine learning practitioners: Instead of picking a single optimizer and tuning its hyperparameters, trying out multiple default setting optimizers and picking the best one should yield competitive results with less computational and tuning choice efforts. The similarity of those two approaches might be due to the fact that optimizers have implicit learning rate schedules and trying out different optimizers is similar to trying out different (well-tested) schedules (Agarwal et al., 2020).

**How much do tuning and schedules help?**    We consider the final performance achieved by varying budgets and schedules to quantify the usefulness of tuning and applying parameter-free schedules (Figure 3). While there is no clear trend for any individual setting (gray lines), in the median we observe that increasing the budget improves performance, albeit with diminishing returns. For

example, using the large budget without any schedule leads to a median relative improvement of the performance of roughly $3.4\%$ compared to the default parameters (without schedule).

Similarly, applying a parameter-free (i.e. untuned) schedule improves median performance. For example, the large tuning budget coupled with a trapezoidal learning rate schedule leads to a median relative improvement of roughly $5.3\%$ compared to the default parameters. However, while these trends hold in the median, their individual effect varies wildly among optimizers and test problems, as is apparent from the noisy structure of the individual lines shown in Figure 3.

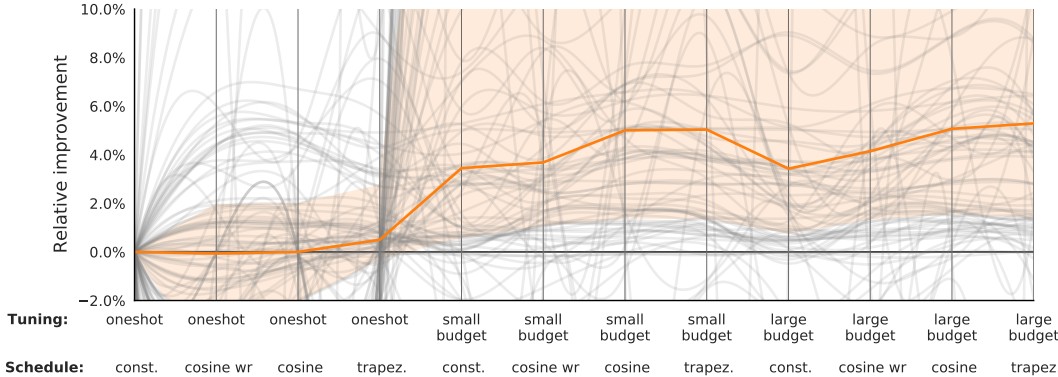

Figure 3: Lines in gray (—, smoothed by cubic splines for visual guidance only) show the relative improvement for a certain tuning and schedule (compared to the *one-shot* tuning without schedule) for all $14$ optimizers on all eight test problems. The median over all lines is plotted in orange (—) with the shaded area (▮) indicating the area between the 25th and 75th percentile.

**Which optimizers work well after tuning?** Figure 4 compares the optimizers' performance across the test problems. There is no single optimizer that dominates its competitors across all tasks. Nevertheless, some optimizers generally perform well, while others vary wildly in their behavior. Further

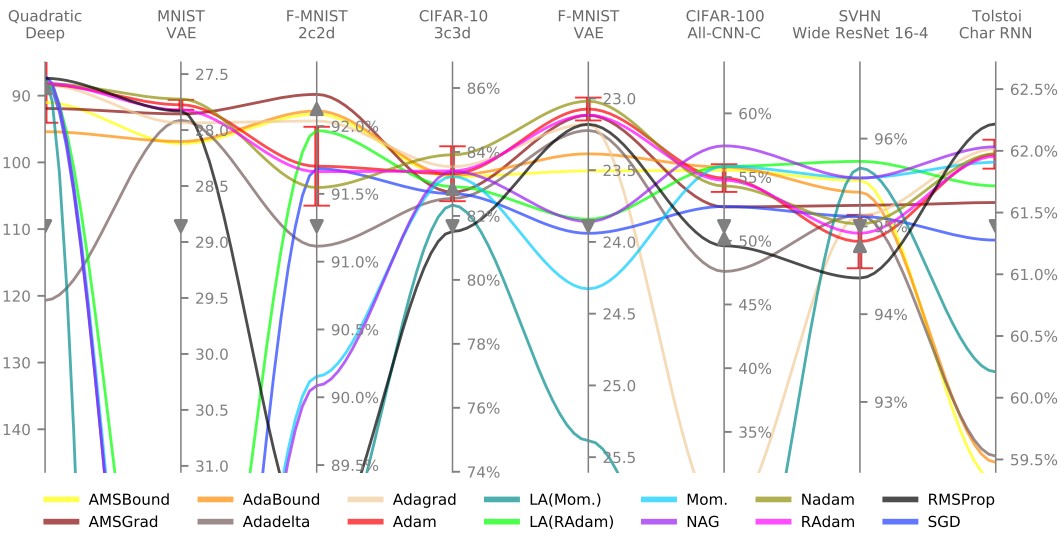

Figure 4: Mean test set performance over $10$ random seeds of all tested optimizers on all eight optimization problems using the *large budget* for tuning and *no learning rate schedule*. One standard deviation for the *tuned* ADAM optimizer is shown with a red error bar (**I**; error bars for other methods omitted for legibility). The performance of *untuned* ADAM (▼) and ADABOUND (▲) are marked for reference. The upper bound of each axis represents the best performance achieved in the benchmark, while the lower bound is chosen in relation to the performance of ADAM with default parameters.

supporting the hypothesis of previous sections, we note that taking the best out of a small set of *un-*

*tuned* optimizers — for example, ADAM and ADABOUND— frequently results in competitive overall performance, even compared to well-tuned optimizers. Combining these runs with a *tuned* version of ADAM (or variants thereof) generally yields competitive results in our benchmark. Nevertheless, achieving (or getting close to) the absolute best performance still requires testing multiple optimizers. Which optimizer wins in the end, though, is problem-dependent: optimizers that achieve top scores on one problem can perform rather badly on other tasks. We note in passing that the individual optimizer rankings can change when considering e.g. a smaller budget or an additional learning rate schedule (see Figures 13 to 15 in the appendix). However, the overall trends described here are consistent.

## 4 LIMITATIONS

Any empirical benchmark has constraints and limitations. Here we highlight some of them and characterize the context within which our results should be considered.

**Generalization of the results**   By using the test problems from DEEPOBS, which span models and data sets of varying complexity, size, and different domains, we aim for generalization. Our results are, despite our best efforts, reflective of not just these setups, but also to the chosen training parameters, the software framework, and further unavoidable choices. The design of our comparisons aims to be close to what an informed practitioners would encounter in practice. It goes without saying that even a carefully curated range of test problems cannot cover all challenges of machine learning or even just deep learning. In particular, our conclusions may not generalize to other types of workloads such as GANs, reinforcement learning, or applications where e.g. memory usage is crucial. Similarly, our benchmark does not cover more large-scale problems such as ImageNet (Deng et al., 2009) or transformer models (Vaswani et al., 2017) for machine translation. Studying, whether there are systematic differences between these types of optimization problems presents an interesting avenue for further research.

We don't consider this study the definitive work on benchmark deep learning optimizers, but rather an important step in the right direction. While our comparison includes many "dimensions" of deep learning optimization, e.g. by considering different problems, tuning budgets, and learning rate schedules, there are many more. To keep the benchmark feasible, we chose to use the fixed L2-regularization and batch size that DEEPOBS suggests for each problem. We also did not include optimization techniques such as weight averaging or ensemble methods as they can be combined with all evaluated optimizers. Future works could study how these techniques interact with different optimization methods. However, to keep our benchmark feasible, we have selected what we believe to be the most important aspects affecting an optimizer comparison. We hope, that our study lays the groundwork so that other works can build on it and analyze these questions.

**Influence of the hyperparameter search strategy**   As noted by, e.g., Choi et al. (2019) and Sivaprasad et al. (2020), the hyperparameter tuning method, its budget, and its search domain, can significantly affect performance. By reporting results from three different hyperparameter optimization budgets (including the tuning-free one-shot setting) we try to quantify the effect of tuning. We argue that our random search process presents a realistic setting for many but certainly not all deep learning practitioners. One may criticize our approach as simplistic, but note that more elaborate schemes, in particular Bayesian optimization, would multiply the number of design decisions (kernels, search utilities, priors, and scales) and thus significantly complicate the analysis.

The individual hyperparameter sampling distributions significantly affect the relative rankings of the optimizers. A badly chosen search space can make tuning next to impossible. Note, though, that this problem is inherited by practitioners. It is arguably an implicit flaw of an optimizer to not come with well-identified search spaces for its hyperparameters and should thus be reflected in a benchmark.

## 5 CONCLUSION

Faced with an avalanche of research to develop new stochastic optimization methods, practitioners are left with the near-impossible task of not just picking a method from this ever-growing list, but also to guess or tune hyperparameters for them, even to continuously tune them during optimization. Despite efforts by the community, there is currently no method that clearly dominates the competition.

We have provided an extensive empirical benchmark of optimization methods for deep learning. It reveals structure in the crowded field of optimization for deep learning: First, although many methods perform competitively, a subset of methods tends to come up near the top across the spectrum of problems. Secondly, tuning helps about as much as trying other optimizers. Our open data set allows many, more technical observations, e.g., that the stability to re-runs is an often overlooked challenge.

Perhaps the most important takeaway from our study is hidden in plain sight: the field is in danger of being drowned by noise. Different optimizers exhibit a surprisingly similar performance distribution compared to a single method that is re-tuned or simply re-run with different random seeds. It is thus questionable how much insight the development of new methods yields, at least if they are conceptually and functionally close to the existing population. We hope that benchmarks like ours can help the community to rise beyond inventing yet another optimizer and to focus on key challenges, such as automatic, inner-loop tuning for truly robust and efficient optimization. We are releasing our data to allow future authors to ensure that their method contributes to such ends.

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

## A  LIST OF OPTIMIZERS AND SCHEDULES CONSIDERED

Table 2: List of optimizers we considered for our benchmark. Note, that this is still far from being a complete list of all existing optimization methods applicable to deep learning, but only a subset, comprising of some of the most popular choices.

| Name | Ref. | Name | Ref. |
|------|------|------|------|
| ACClip | (Zhang et al., 2020) | LARS | (You et al., 2017) |
| ADAHESSIAN | (Yao et al., 2020) | LookAhead | (Zhang et al., 2019) |
| ALI-G | (Berrada et al., 2020) | M-SVAG | (Balles & Hennig, 2018) |
| AMSBound | (Luo et al., 2019) | MTAdam | (Malkiel & Wolf, 2020) |
| AMSGrad | (Reddi et al., 2018) | Nesterov | (Nesterov, 1983) |
| ARSG | (Chen et al., 2019b) | Polyak | (Polyak, 1964) |
| AcceleGrad | (Levy et al., 2018) | MVRC-1 | (Chen & Zhou, 2020) |
| AdaAlter | (Xie et al., 2019) | MVRC-2 | (Chen & Zhou, 2020) |
| AdaBatch | (Devarakonda et al., 2017) | NAMSB | (Chen et al., 2019b) |
| AdaBayes | (Aitchison, 2020) | NAMSG | (Chen et al., 2019b) |
| AdaBayes-SS | (Aitchison, 2020) | ND-Adam | (Zhang et al., 2017) |
| AdaBlock | (Yun et al., 2019) | Nadam | (Dozat, 2016) |
| AdaBound | (Luo et al., 2019) | Noisy Adam | (Zhang et al., 2018) |
| AdaComp | (Chen et al., 2018) | Noisy K-FAC | (Zhang et al., 2018) |
| AdaFTRL | (Orabona & Pál, 2015) | NosAdam | (Huang et al., 2019) |
| AdaFix | (Bae et al., 2019) | Novograd | (Ginsburg et al., 2019) |
| AdaFom | (Chen et al., 2019a) | PAGE | (Li et al., 2020b) |
| AdaLoss | (Teixeira et al., 2019) | PAL | (Mutschler & Zell, 2020) |
| AdaMax | (Kingma & Ba, 2015) | Padam | (Chen et al., 2020) |
| AdaMod | (Ding et al., 2019) | PolyAdam | (Orvieto et al., 2019) |
| AdaScale | (Johnson et al., 2020) | PowerSGD | (Vogels et al., 2019) |
| AdaSGD | (Wang & Wiens, 2020) | PowerSGDM | (Vogels et al., 2019) |
| AdaShift | (Zhou et al., 2019) | ProbLS | (Mahsereci, 2018) |
| AdaSqrt | (Hu et al., 2019) | PStorm | (Xu, 2020) |
| AdaX | (Li et al., 2020a) | QHAdam | (Ma & Yarats, 2019) |
| AdaX-W | (Li et al., 2020a) | QHM | (Ma & Yarats, 2019) |
| Adadelta | (Zeiler, 2012) | RAdam | (Liu et al., 2020) |
| Adagrad | (Duchi et al., 2011) | RMSProp | (Tieleman & Hinton, 2012) |
| Adam | (Kingma & Ba, 2015) | RMSterov | (Choi et al., 2019) |
| AdamAL | (Tao et al., 2019) | Ranger | (Wright, 2020b) |
| AdamNC | (Reddi et al., 2018) | RangerLars | (Grankin, 2020) |
| AdamP | (Heo et al., 2020) | SAMSGrad | (Tong et al., 2019) |
| AdamT | (Zhou et al., 2020) | SAdam | (Wang et al., 2020b) |
| AdamW | (Loshchilov & Hutter, 2019) | SC-Adagrad | (Mukkamala & Hein, 2017) |
| AdamX | (Tran & Phong, 2019) | SC-RMSProp | (Mukkamala & Hein, 2017) |
| Adathm | (Sun et al., 2019) | SDProp | (Ida et al., 2017) |
| ArmijoLS | (Vaswani et al., 2019) | SGD | (Robbins & Monro, 1951) |
| AvaGrad | (Savarese et al., 2019) | SGD-BB | (Tan et al., 2016) |
| BAdam | (Salas et al., 2018) | SGD-G2 | (Ayadi & Turinici, 2020) |
| BGAdam | (Bai & Zhang, 2019) | SGDM | (Liu & Luo, 2020) |
| BRMSProp | (Aitchison, 2020) | SGDP | (Heo et al., 2020) |
| BSGD | (Hu et al., 2020) | SGDR | (Loshchilov & Hutter, 2017) |
| C-ADAM | (Tutunov et al., 2020) | SHAdagrad | (Huang et al., 2020) |
| CProp | (Preechakul & Kijsirikul, 2019) | SKQN | (Yang et al., 2020) |
| Cool Momentum | (Borysenko & Byshkin, 2020) | S4QN | (Yang et al., 2020) |
| Compositional ADAM | (Tutunov et al., 2020) | SNGM | (Zhao et al., 2020) |
| Curveball | (Henriques et al., 2019) | SRSGD | (Wang et al., 2020a) |
| Dadam | (Nazari et al., 2019) | SWATS | (Keskar & Socher, 2017) |
| DeepMemory | (Wright, 2020a) | SWNTS | (Chen et al., 2019c) |
| DiffGrad | (Dubey et al., 2020) | SADAM | (Tong et al., 2019) |
| Eve | (Longhi, 2017) | Shampoo | (Anil et al., 2020; Gupta et al., 2018) |
| Gadam | (Zhang & Gouza, 2018) | SignAdam++ | (Wang et al., 2019a) |
| GOLS-I | (Kafka & Wilke, 2019) | SignSGD | (Bernstein et al., 2018) |
| Gravilon | (Kelterborn et al., 2020) | SoftAdam | (Fetterman et al., 2019) |
| HAdam | (Jiang et al., 2019) | S-SGD | (Sung et al., 2020) |
| HyperAdam | (Wang et al., 2019b) | TAdam | (Ilboudo et al., 2020) |
| K-BFGS | (Goldfarb et al., 2020) | VAdam | (Khan et al., 2018) |
| K-BFGS(L) | (Goldfarb et al., 2020) | VR-SGD | (Shang et al., 2020) |
| KFAC | (Martens & Grosse, 2015) | WNGrad | (Wu et al., 2018) |
| KFLR | (Botev et al., 2017) | YellowFin | (Zhang & Mitliagkas, 2019) |
| KFRA | (Botev et al., 2017) | Yogi | (Zaheer et al., 2018) |
| L4Adam | (Rolínek & Martius, 2018) | vSGD-b | (Schaul et al., 2013) |
| L4Momentum | (Rolínek & Martius, 2018) | vSGD-fd | (Schaul & LeCun, 2013) |
| LAMB | (You et al., 2020) | vSGD-g | (Schaul et al., 2013) |
| LaProp | (Ziyin et al., 2020) | vSGD-l | (Schaul et al., 2013) |

Table 3: Overview of commonly used parameter schedules. Note, while we list the schedules parameters, it isn't clearly defined what aspects of a schedule are (tunable) parameters and what is a-priori fixed. In this column, $\alpha_0$ denotes the initial learning rate, $\alpha_{lo}$ and $\alpha_{up}$ the lower and upper bound, $\Delta t$ indicates an epoch count at which to switch decay styles, $k$ denotes a decaying factor.

| Name | | Ref. | Illustration | Parameters |
|---|---|---|---|---|
| Constant | | | | $\alpha_0$ |
| Step Decay | constant factor | | | $\alpha_0, \Delta t_1, \ldots, k$ |
| | multi-step | | | $\alpha_0, \Delta t_1, \ldots, k_1, \ldots$ |
| Smooth Decay | linear decay | e.g. (Goodfellow et al., 2016) | | $\alpha_0, (\Delta t, \alpha_{lo})$ |
| | polynomial decay | | | $\alpha_0, k, (\alpha_{lo})$ |
| | exponential decay | | | $\alpha_0, k, (\alpha_{lo})$ |
| | inverse time decay | e.g. (Bottou, 2012) | | $\alpha_0, k, (\alpha_{lo})$ |
| | cosine decay | (Loshchilov & Hutter, 2017) | | $\alpha_0, (\alpha_{lo})$ |
| | linear cosine decay | (Bello et al., 2017) | | $\alpha_0, (\alpha_{lo})$ |
| Cyclical | triangular | (Smith, 2017) | | $\alpha_{lo}, \alpha_{up}, \Delta t$ |
| | triangular + decay | (Smith, 2017) | | $\alpha_{lo}, \alpha_{up}, \Delta t, k$ |
| | triangular + exponential decay | (Smith, 2017) | | $\alpha_{lo}, \alpha_{up}, \Delta t$ |
| | cosine + warm restarts | (Loshchilov & Hutter, 2017) | | $\alpha_{up}, \Delta t, (\alpha_{lo})$ |
| | cosine + warm restarts + decay | (Loshchilov & Hutter, 2017) | | $\alpha_{up}, \Delta t, k, (\alpha_{lo})$ |
| Warmup | constant warmup | e.g. (He et al., 2016) | | $\alpha_{lo}, \alpha_0, \Delta t$ |
| | gradual warmup | (Goyal et al., 2017) | | $\alpha_0, \Delta t, (\alpha_{lo})$ |
| | gradual warmup + multi-step decay | (Goyal et al., 2017) | | $\alpha_0, \Delta t, \Delta t_{steps}, k_1, \ldots, (\alpha_{lo})$ |
| | gradual warmup + step number decay | (Vaswani et al., 2017) | | $\alpha_0, \Delta t, (\alpha_{lo})$ |
| | slanted triangular | (Howard & Ruder, 2018) | | $\alpha_0, \Delta t, (\alpha_{lo})$ |
| | long trapezoid | (Xing et al., 2018) | | $\alpha_0, \Delta t_{up}, \Delta t_{down}, (\alpha_{lo})$ |
| Super-Convergence | 1cycle | (Smith & Topin, 2017) | | $\alpha_{up}, \Delta t, \Delta t_{cutoff}, (\alpha_{lo})$ |

## B    LIST OF OPTIMIZERS SELECTED

Table 4: Selected optimizers for our benchmarking process with their respective color, hyperparameters, default values, tuning distributions and scheduled hyperparameters. Here, $\mathcal{LU}(\cdot, \cdot)$ denotes the log-uniform distribution while $\mathcal{U}\{\cdot, \cdot\}$ denotes the discrete uniform distribution.

| Optimizer | Ref. | Parameters | Default | Tuning Distribution | Scheduled |
|---|---|---|---|---|---|
| ● AMSBound | (Luo et al., 2019) | $\alpha$ | $10^{-3}$ | $\mathcal{LU}(10^{-4}, 1)$ | ✓ |
|  |  | $\alpha_l$ | 0.1 | $\mathcal{LU}(10^{-3}, 0.5)$ |  |
|  |  | $\beta_1$ | 0.9 | $\mathcal{LU}(0.5, 0.999)$ |  |
|  |  | $\beta_2$ | 0.999 | $\mathcal{LU}(0.8, 0.999)$ |  |
|  |  | $\gamma$ | $10^{-3}$ | $\mathcal{LU}(10^{-4}, 10^{-1})$ |  |
|  |  | $\varepsilon$ | $10^{-8}$ | ✗ |  |
| ● AMSGrad | (Reddi et al., 2018) | $\alpha$ | $10^{-2}$ | $\mathcal{LU}(10^{-4}, 1)$ | ✓ |
|  |  | $\beta_1$ | 0.9 | $\mathcal{LU}(0.5, 0.999)$ |  |
|  |  | $\beta_2$ | 0.999 | $\mathcal{LU}(0.8, 0.999)$ |  |
|  |  | $\varepsilon$ | $10^{-8}$ | ✗ |  |
| ● AdaBound | (Luo et al., 2019) | $\alpha$ | $10^{-3}$ | $\mathcal{LU}(10^{-4}, 1)$ | ✓ |
|  |  | $\alpha_l$ | 0.1 | $\mathcal{LU}(10^{-3}, 0.5)$ |  |
|  |  | $\beta_1$ | 0.9 | $\mathcal{LU}(0.5, 0.999)$ |  |
|  |  | $\beta_2$ | 0.999 | $\mathcal{LU}(0.8, 0.999)$ |  |
|  |  | $\gamma$ | $10^{-3}$ | $\mathcal{LU}(10^{-4}, 10^{-1})$ |  |
| ● Adadelta | (Zeiler, 2012) | $\alpha$ | $10^{-3}$ | $\mathcal{LU}(10^{-4}, 1)$ | ✓ |
|  |  | $\varepsilon$ | $10^{-8}$ | ✗ |  |
|  |  | $1 - \rho$ | 0.95 | $\mathcal{LU}(10^{-4}, 1)$ |  |
| ● Adagrad | (Duchi et al., 2011) | $\alpha$ | $10^{-2}$ | $\mathcal{LU}(10^{-4}, 1)$ | ✓ |
|  |  | $\varepsilon$ | $10^{-7}$ | ✗ |  |
| ● Adam | (Kingma & Ba, 2015) | $\alpha$ | $10^{-3}$ | $\mathcal{LU}(10^{-4}, 1)$ | ✓ |
|  |  | $\beta_1$ | 0.9 | $\mathcal{LU}(0.5, 0.999)$ |  |
|  |  | $\beta_2$ | 0.999 | $\mathcal{LU}(0.8, 0.999)$ |  |
|  |  | $\varepsilon$ | $10^{-8}$ | ✗ |  |
| ● Lookahead Momentum abbr. LA(Mom.) | (Zhang et al., 2019) | $\alpha$ | 0.5 | $\mathcal{LU}(10^{-4}, 1)$ |  |
|  |  | $\alpha_f$ | $10^{-2}$ | $\mathcal{LU}(10^{-4}, 1)$ | ✓ |
|  |  | $k$ | 5 | $\mathcal{U}\{1, 20\}$ |  |
|  |  | $1 - \rho$ | 0.99 | $\mathcal{LU}(10^{-4}, 1)$ |  |
| ● Lookahead RAdam abbr. LA(RAdam) | (Zhang et al., 2019) | $\alpha$ | 0.5 | $\mathcal{LU}(10^{-4}, 1)$ |  |
|  |  | $\alpha_f$ | $10^{-3}$ | $\mathcal{LU}(1e-4, 1)$ | ✓ |
|  |  | $\beta_1$ | 0.9 | $\mathcal{LU}(0.5, 0.999)$ |  |
|  |  | $\beta_2$ | 0.999 | $\mathcal{LU}(0.8, 0.999)$ |  |
|  |  | $\varepsilon$ | $10^{-7}$ | ✗ |  |
|  |  | $k$ | 5 | $\mathcal{U}\{1, 20\}$ |  |
| ● Momentum | (Polyak, 1964) | $\alpha$ | $10^{-2}$ | $\mathcal{LU}(10^{-4}, 1)$ | ✓ |
|  |  | $1 - \rho$ | 0.99 | $\mathcal{LU}(10^{-4}, 1)$ |  |
| ● NAG | (Nesterov, 1983) | $\alpha$ | $10^{-2}$ | $\mathcal{LU}(10^{-4}, 1)$ | ✓ |
|  |  | $1 - \rho$ | 0.99 | $\mathcal{LU}(10^{-4}, 1)$ |  |
| ● NAdam | (Dozat, 2016) | $\alpha$ | $10^{-3}$ | $\mathcal{LU}(10^{-4}, 1)$ | ✓ |
|  |  | $\beta_1$ | 0.9 | $\mathcal{LU}(0.5, 0.999)$ |  |
|  |  | $\beta_2$ | 0.999 | $\mathcal{LU}(0.8, 0.999)$ |  |
|  |  | $\varepsilon$ | $10^{-7}$ | ✗ |  |
| ● RAdam | (Liu et al., 2020) | $\alpha$ | $10^{-3}$ | $\mathcal{LU}(10^{-4}, 1)$ | ✓ |
|  |  | $\beta_1$ | 0.9 | $\mathcal{LU}(0.5, 0.999)$ |  |
|  |  | $\beta_2$ | 0.999 | $\mathcal{LU}(0.8, 0.999)$ |  |
|  |  | $\varepsilon$ | $10^{-7}$ | ✗ |  |
| ● RMSProp | (Tieleman & Hinton, 2012) | $\alpha$ | $10^{-3}$ | $\mathcal{LU}(10^{-4}, 1)$ | ✓ |
|  |  | $\varepsilon$ | $10^{-10}$ | ✗ |  |
|  |  | $1 - \rho$ | 0.9 | $\mathcal{LU}(10^{-4}, 1)$ |  |
| ● SGD | (Robbins & Monro, 1951) | $\alpha$ | $10^{-2}$ | $\mathcal{LU}(10^{-4}, 1)$ | ✓ |

## C ROBUSTNESS TO RANDOM SEEDS

Data subsampling, random weight initialization, dropout and other aspects of deep learning introduce stochasticity to the training process. As such, judging the performance of an optimizer on a single run may be misleading due to random fluctuations. In our benchmark we use 10 different seeds of the final setting for each budget in order to judge the stability of the optimizer and the results. However, to keep the magnitude of this benchmark feasible, we only use a single seed while tuning, analogously to how a single user would progress. This means that our tuning process can sometimes choose hyperparameter settings which might not even converge for seeds other than the one used for tuning.

Figure 5 illustrates this behavior on an example problem where we used 10 seeds throughout a tuning process using grid search. The figure shows that in the beginning performance increases when increasing the learning rate, followed by an area were it sometimes works but other times diverges. Picking hyperparameters from this "danger zone" can lead to unstable results. In this case, where we only consider the learning rate, it is clear that decreasing the learning rate a bit to get away from this "danger zone" would lead to a more stable, but equally well-performing algorithm. In more complicated cases, however, we are unable to use a simple heuristic such as this. This might be the case, for example, when tuning multiple hyperparameters or when the effect of the hyperparameter on the performance is less straight forward. Thus, this is a problem not created by improperly using the tuning method, but by an unstable optimization method.

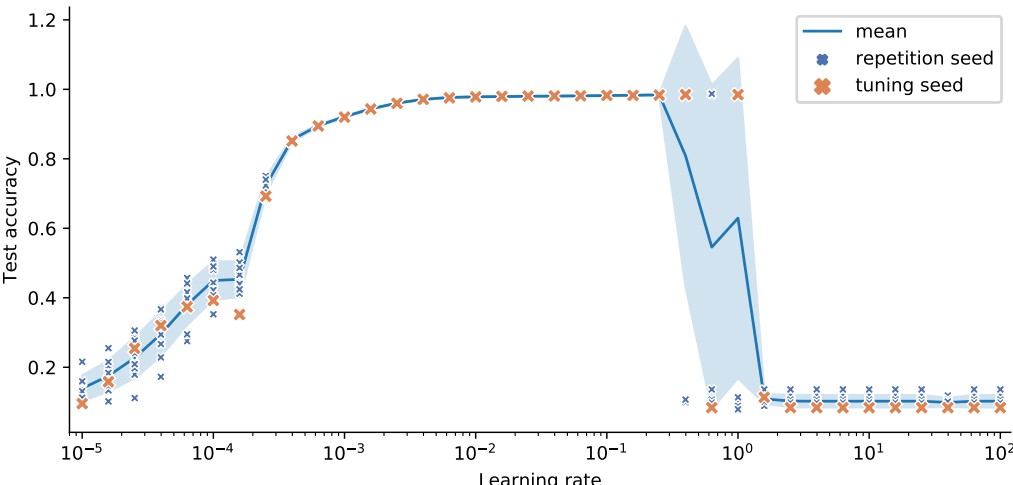

Figure 5: Performance of SGD on a simple multilayer perceptron. For each learning rate, markers in orange (✖) show the initial seed which would be used for tuning, blue markers (✖) illustrate nine additional seeds with otherwise unchanged settings. The mean over all seeds is plotted as a blue line (—), showing one standard deviation as a shaded area (⬤).

In our benchmark, we observe in total 49 divergent seeds for the small budget and 56 for the large budget, or roughly 1% of the runs in each budget. Most of them occur when using SGD (23 and 18 cases for the small and large budget respectively), MOMENTUM (13 and 17 cases for the small and large budget respectively) or NAG (7 and 12 cases for the small and large budget respectively), which might indicate that adaptive methods are less prone to this kind of behavior. For the small budget tuning, none of these cases occur when using a constant schedule (4 for the large budget), and most of them occur when using the *cosine with warm restarts* schedule (27 and 25 cases for the small and large budget respectively). However, as our data on diverging seeds is very limited, it is not conclusive enough to draw solid conclusions.

# D RE-TUNING EXPERIMENTS

In order to test the stability of our benchmark and especially the tuning method, we selected two optimizers in our benchmark and re-tuned them on all test problems a second time. We used completely independent random seeds for both tuning and the 10 repetitions with the final setting. Figure 6 and Figure 7 show the distribution of all 10 random seeds for both the original tuning as well as the re-tuning runs for RMSPROP and ADADELTA. It is evident, that re-tuning results in a shift of this distribution, since small (stochastic) changes during tuning can result in a different chosen hyperparameter setting.

These differences also highlight how crucial it is to look at multiple test problems. Individually, small changes, such as re-doing the tuning with different seeds can lead to optimization methods changing rankings. However, they tend to average out when looking at an unbiased list of multiple problems. These results also further supports the statement made in Section 3 that there is no optimization method clearly domination the competition, as small performance margins might vanish when re-tuning.

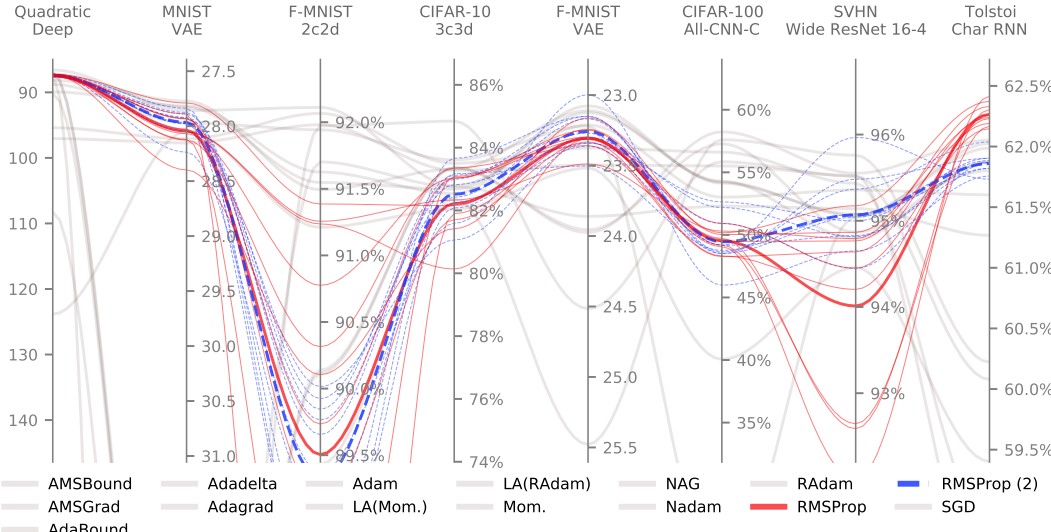

Figure 6: Mean test set performance of all 10 seeds of RMSPROP (—) on all eight optimization problems using the *small budget* for tuning and *no learning rate schedule*. The mean is shown with a thicker line. We repeated the full tuning process on all eight test problems using different random seeds, which is shown in dashed lines blue (- -). The mean performance of all other optimizers is shown in transparent gray lines.

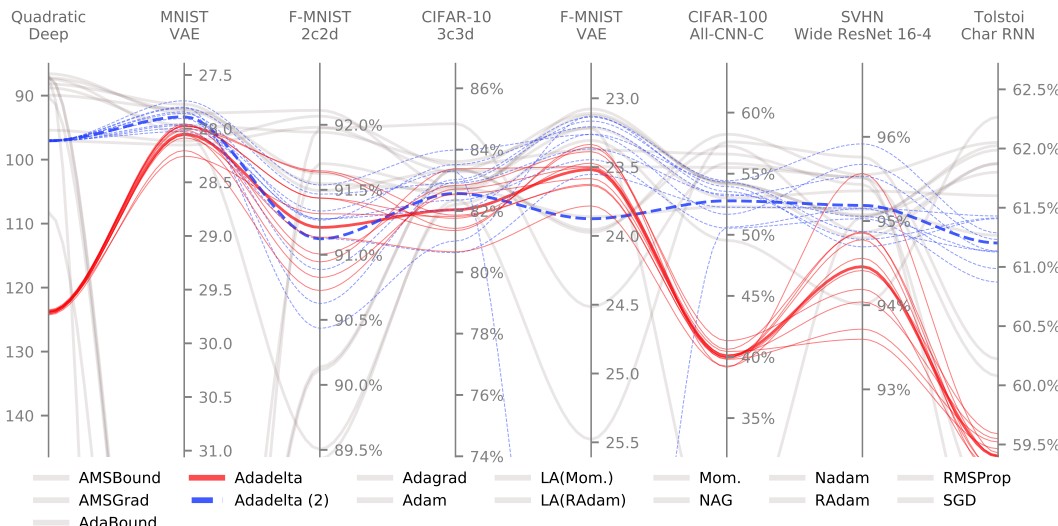

Figure 7: Mean test set performance of all 10 seeds of ADADELTA (——) on all eight optimization problems using the *small budget* for tuning and *no learning rate schedule*. The mean is shown with a thicker line. We repeated the full tuning process on all eight test problems using different random seeds, which is shown in dashed lines blue (- -). The mean performance of all other optimizers is shown in transparent gray lines.

# E    LIST OF SCHEDULES SELECTED

The schedules selected for our benchmark are illustrated in Figure 8. All learning rate schedules are multiplied by the initial learning rate found via tuning or picked as the default choice.

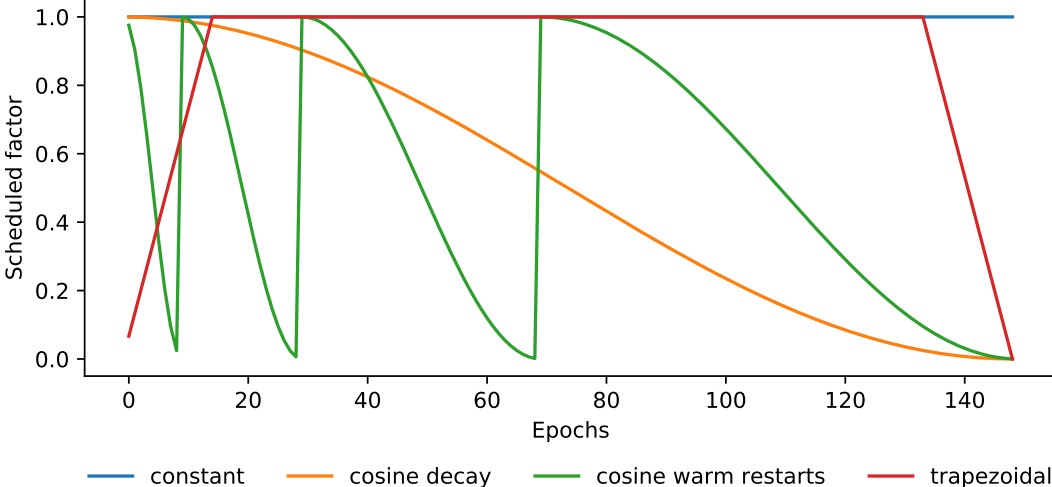

Figure 8: Illustration of the selected learning rate schedules for a training duration of $150$ epochs.

We use a *cosine decay* (Loshchilov & Hutter, 2017) that starts at $1$ and decays in the form of a half period of a cosine to $0$. As an example of a cyclical learning rate schedule, we test a *cosine with warm restarts* schedule with a cycle length $\Delta t = 10$ which increases by a factor of $2$ after each cycle without any discount factor. Depending on the number of epochs we train our model, it is possible that training stops shortly after one of those warm restarts. Since performance typically declines shortly after increasing the learning rate, we don't report the final performance for this schedule, but instead the performance achieved after the last complete period (just before the next restart). This approach is suggested by the original work of Loshchilov & Hutter (2017). However, we still use the final performance while tuning.

A representation of a schedule including warm-up is the *trapezoidal* schedule from Xing et al. (2018). For our benchmark we set a warm-up and cool-down period of $1/10$ the training time.

# F IMPROVEMENT AFTER TUNING

When looking at Figure 2, one might realize that few diagonal entries contain negative values. Since diagonal entries reflect the intra-optimizer performance change when tuning on the respective task, this might feel quite counterintuitive at first. *In theory*, this can occur if the respective tuning distributions is chosen poorly, the tuning randomness simply got "unlucky", or we observe significantly worse results for our additional seeds (see Figure 5).

If we compare Figures 9 and 10 to Figures 11 and 12 we can see most negative diagonal entries vanish or at least diminish in magnitude. For the latter two figures we allow for more tuning runs and only consider the seed that has been used for this tuning process. The fact that the effect of negative diagonal entries reduces is an indication that they mostly result from the two latter reasons mentioned.

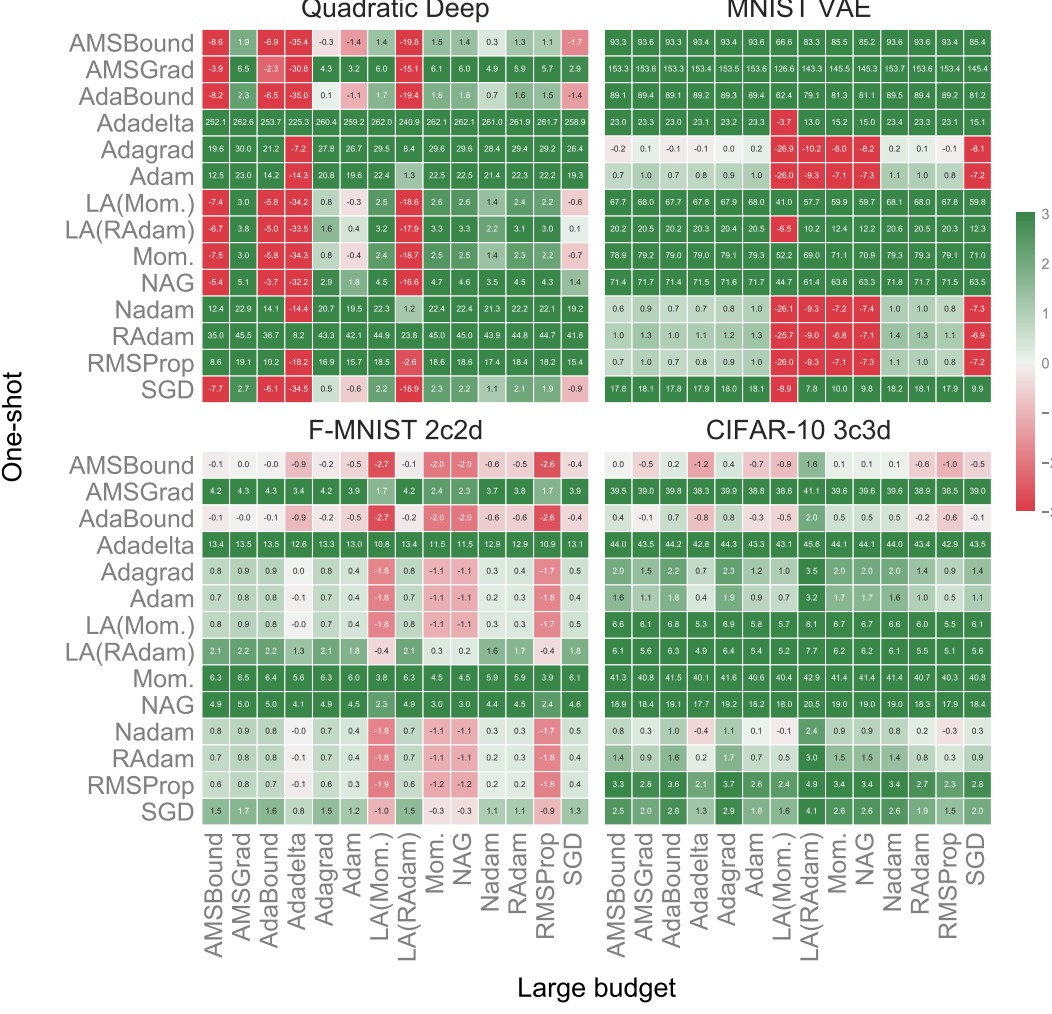

Figure 9: The absolute test set performance improvement after switching from any untuned optimizer (*y*-axis, *one-shot*) to any tuned optimizer (*x*-axis, *small budget*) as an average over 10 random seeds for the *constant* schedule. This is a detailed version of Figure 2 in the main text showing the first four problems.

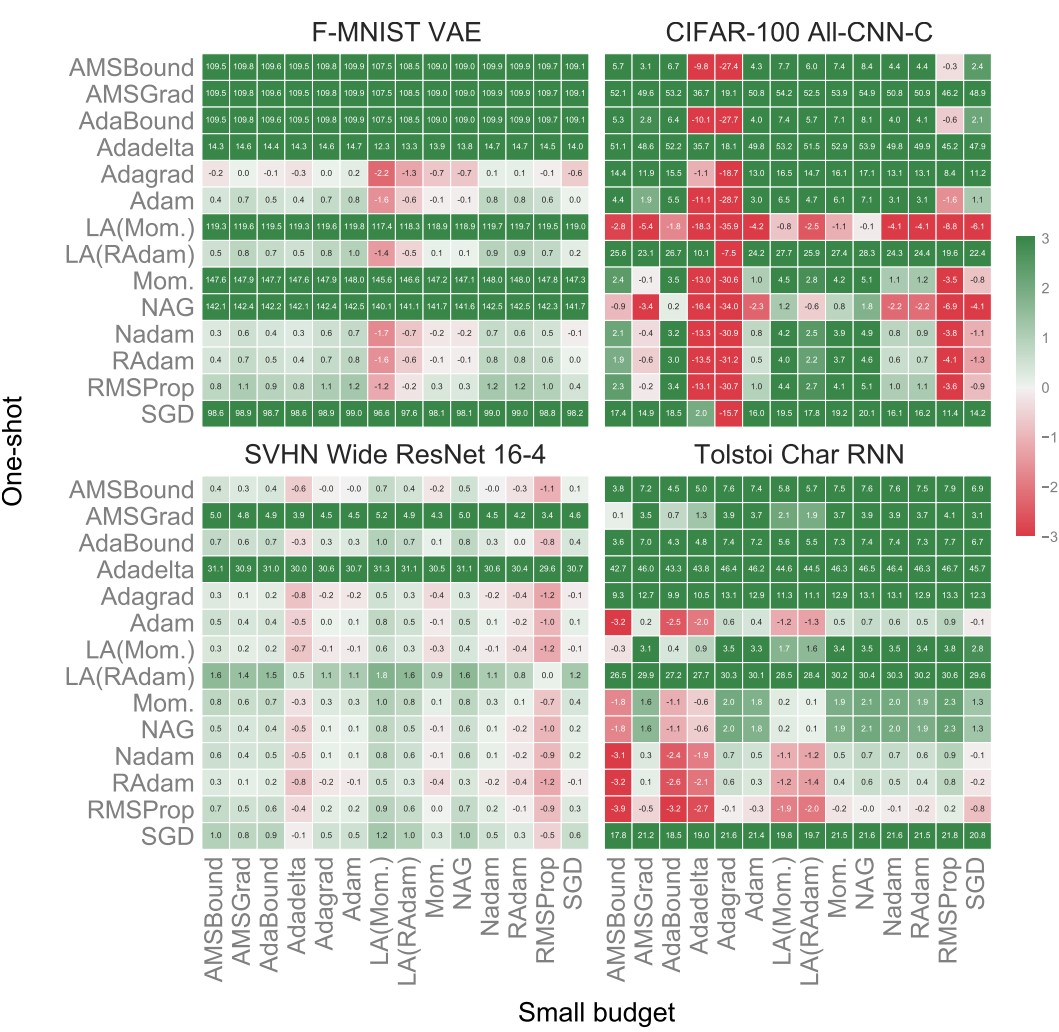

Figure 10: The absolute test set performance improvement after switching from any untuned optimizer ($y$-axis, *one-shot*) to any tuned optimizer ($x$-axis, *small budget*) as an average over 10 random seeds for the *constant* schedule. This is a detailed version of Figure 2 in the main text showing the last four problems.

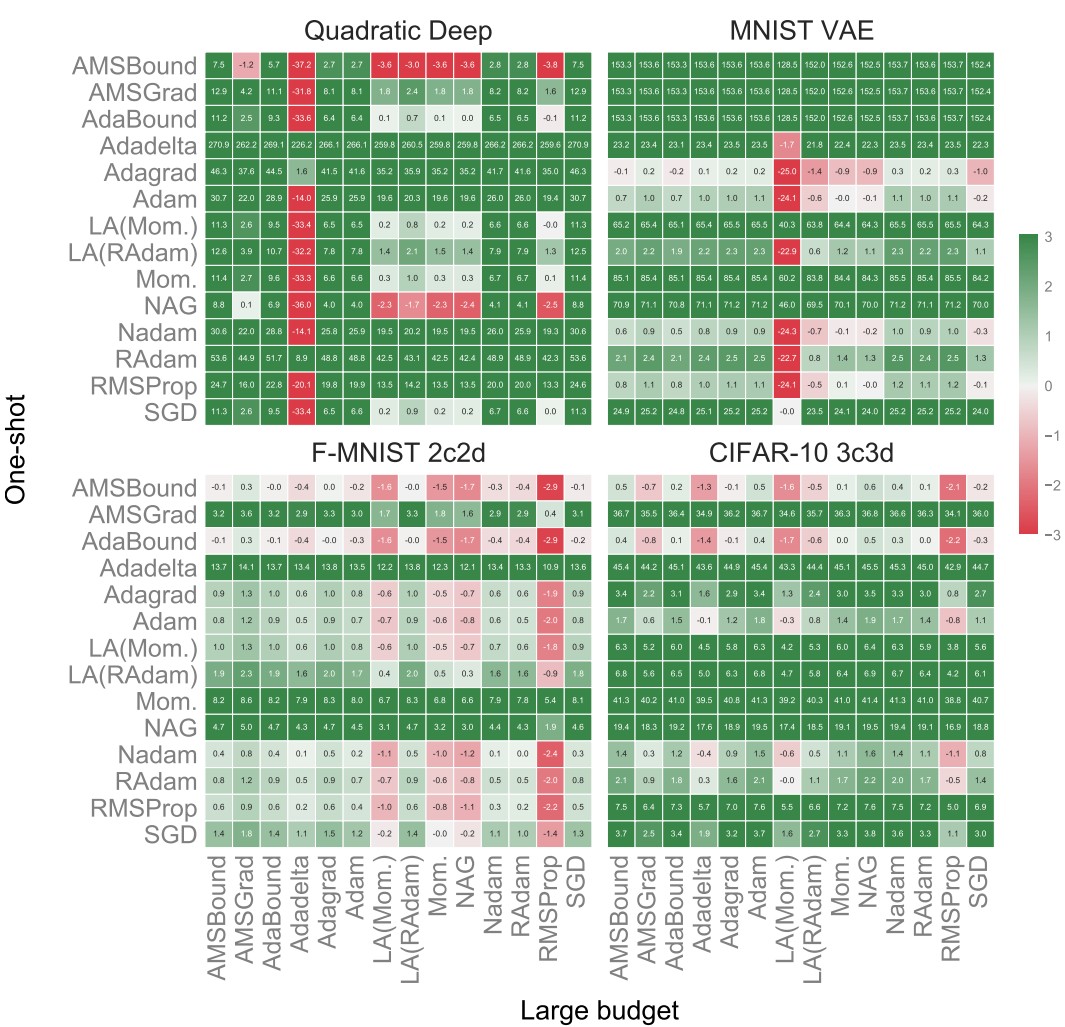

Figure 11: The absolute test set performance improvement after switching from any untuned optimizer (*y*-axis, *one-shot*) to any tuned optimizer (*x*-axis, *large budget*) for the *constant* schedule. This is structurally the same plot as Figure 9 but comparing to the *large budget* and only considering the seed that has been used for tuning.

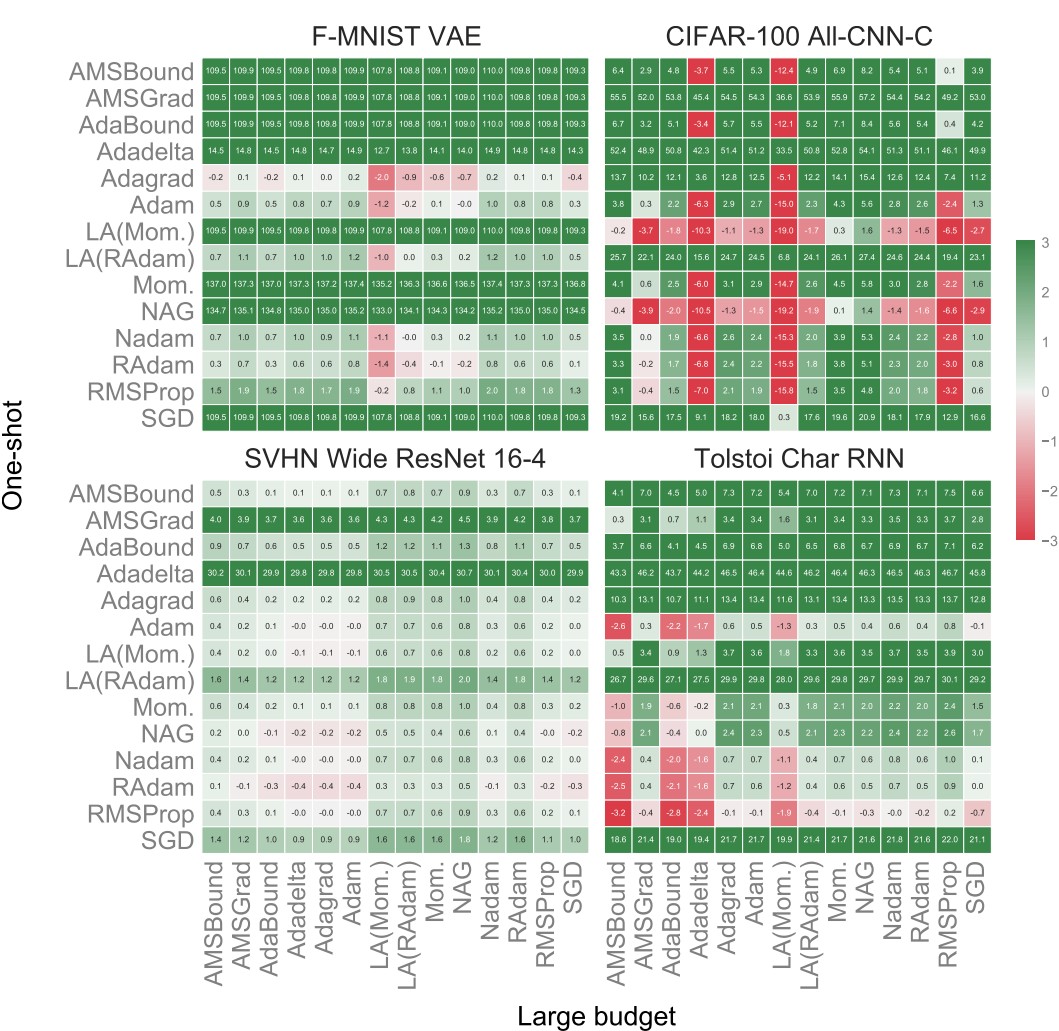

Figure 12: The absolute test set performance improvement after switching from any untuned optimizer (*y*-axis, *one-shot*) to any tuned optimizer (*x*-axis, *large budget*) for the *constant* schedule. This is structurally the same plot as Figure 10 but comparing to the *large budget* and only considering the seed that has been used for tuning.

## G OPTIMIZER PERFORMANCE ACROSS TEST PROBLEMS

Similarly to Figure 4, we show the corresponding plots for the *small budget* with *no learning rate schedule* in Figure 13 and the *large budget* with the *cosine* and *trapezoidal learning rate schedule* in Figures 14 and 15. Additionally, in Figure 16 we show the same setting as Figure 4 but showing the training loss instead of the test loss/accuracy.

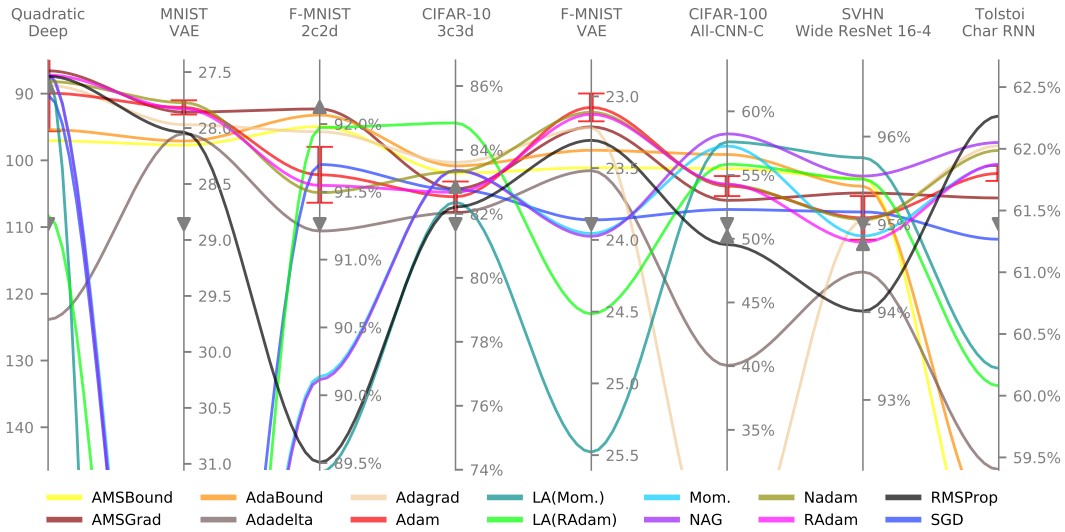

Figure 13: Mean test set performance over 10 random seeds of all tested optimizers on all eight optimization problems using the *small budget* for tuning and *no learning rate schedule*. One standard deviation for the tuned ADAM optimizer is shown with a red error bar (I). The performance of the untuned versions of ADAM (▼) and ADABOUND (▲) are marked for reference. Note, the upper bound of each axis represents the best performance achieved in the benchmark, while the lower bound is chosen in relation to the performance of ADAM with default parameters.

The high-level trends mentioned in Section 3 also hold for the smaller tuning budget in Figure 13. Namely, taking the winning optimizer for several untuned algorithms (here marked for ADAM and ADABOUND) will result in a decent performance in most test problems with much less effort. Adding a tuned version ADAM (or variants thereof) to this selection would result in a very competitive performance. The absolute top-performance however, is achieved by changing optimizers across different test problems.

Note, although the *large budget* is a true superset of the *small budget* it is not given that it will always perform better. Our tuning procedure guarantees that the *validation* performance on the seed that has been used for tuning is as least as good on the large budget than on the small budget. But due to averaging over multiple seeds and reporting *test* performance instead of *validation* performance, this hierarchy is no longer guaranteed. We discuss the possible effects of averaging over multiple seeds further in Appendix C.

The same high-level trends also emerge when considering the *cosine* or *trapezoidal learning rate schedule* in Figures 14 and 15. We can also see that the top performance in general increase when adding a schedule (cf. Figure 4 and Figure 15).

Comparing Figure 4 and Figure 16 we can assess the generalization performance of the optimization method not only to an unseen test set, but also to a different performance metric (accuracy instead of loss). Again, the overall picture of varying performance across different test problems remains consistent when considering the training loss performance. Similarly to the figures showing test set performance we cannot identify a clear winner, although ADAM ands its variants, such as RADAM perform near the top consistently. Note that while Figure 16 shows the training loss, the optimizers have still be tuned to achieve the best validation performance (i.e. accuracy if available, else the loss).

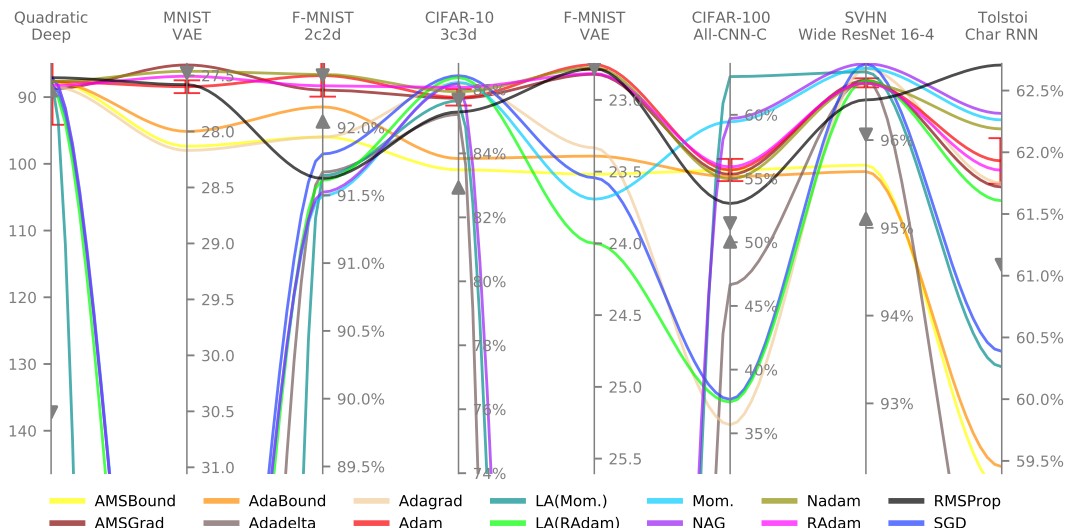

Figure 14: Mean test set performance over 10 random seeds of all tested optimizers on all eight optimization problems using the *large budget* for tuning and the *cosine learning rate schedule*. One standard deviation for the tuned ADAM optimizer is shown with a red error bar (**I**). The performance of the untuned versions of ADAM (▼) and ADABOUND (▲) are marked for reference (this time with the *cosine* learning rate schedule). Note, the upper bound of each axis represents the best performance achieved in the benchmark, while the lower bound is chosen in relation to the performance of ADAM with default parameters (and no schedule).

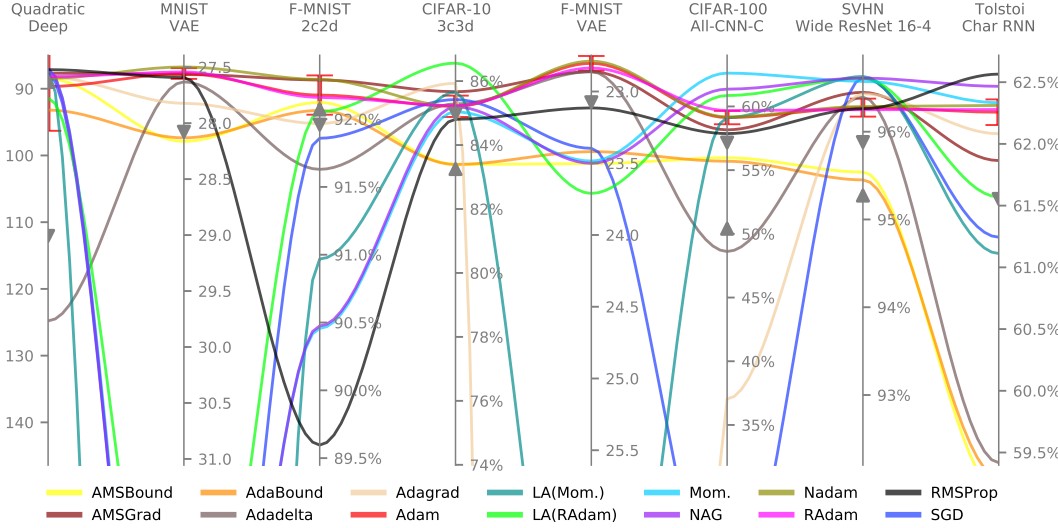

Figure 15: Mean test set performance over 10 random seeds of all tested optimizers on all eight optimization problems using the *large budget* for tuning and the *trapezoidal learning rate schedule*. One standard deviation for the tuned ADAM optimizer is shown with a red error bar (**I**). The performance of the untuned versions of ADAM (▼) and ADABOUND (▲) are marked for reference (this time with the *trapezoidal* learning rate schedule). Note, the upper bound of each axis represents the best performance achieved in the benchmark, while the lower bound is chosen in relation to the performance of ADAM with default parameters (and no schedule).

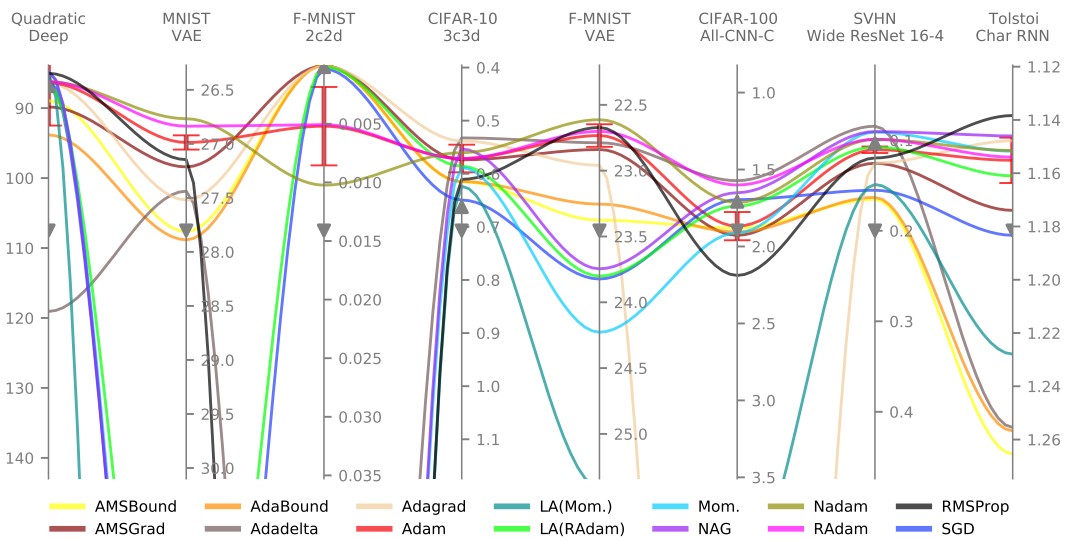

Figure 16: Mean *training* loss performance over 10 random seeds of all tested optimizers on all eight optimization problems using the *large budget* for tuning and *no learning rate schedule*. One standard deviation for the tuned ADAM optimizer is shown with a red error bar (I). The performance of the untuned versions of ADAM (▼) and ADABOUND (▲) are marked for reference. Note, the upper bound of each axis represents the best performance achieved in the benchmark, while the lower bound is chosen in relation to the performance of ADAM with default parameters (and no schedule).

## H  TABULAR VERSION

Table 5: Tabular version of Figure 4. Mean test set performance and standard deviation over 10 random seeds of all tested optimizers on all eight optimization problems using the *large budget* for tuning and *no learning rate schedule*. For comprehensability, mean and standard deviation are rounded.

| Optimizer | Quadratic Deep | MNIST VAE | F-MNIST 2c2d | CIFAR-10 3c3d | F-MNIST VAE | CIFAR-100 | SVHN | Tolstoi |
|---|---|---|---|---|---|---|---|---|
| ● AMSBound | 91.02 ± 7.9 | 28.12 ± 0.1 | 92.09 ± 0.2 | 83.22 ± 1.0 | 23.51 ± 0.1 | 55.54 ± 0.9 | 95.52 ± 0.2 | 59.33 ± 0.4 |
| ● AMSGrad | 91.92 ± 6.2 | 27.86 ± 0.1 | **92.23 ± 0.1** | 82.75 ± 0.6 | 23.12 ± 0.1 | 52.65 ± 1.2 | 95.24 ± 0.2 | 61.58 ± 0.1 |
| ● AdaBound | 95.39 ± 9.1 | 28.10 ± 0.1 | 92.11 ± 0.2 | 83.28 ± 1.0 | 23.39 ± 0.1 | 55.78 ± 0.9 | 95.39 ± 0.2 | 59.48 ± 0.3 |
| ● Adadelta | 120.7 ± 0.2 | 27.91 ± 0.0 | 91.11 ± 0.3 | 82.55 ± 0.9 | 23.22 ± 0.1 | 47.59 ± 0.8 | 95.13 ± 0.3 | 59.53 ± 0.2 |
| ● Adagrad | 88.39 ± 5.7 | 27.94 ± 0.3 | 92.04 ± 0.2 | 83.54 ± 1.0 | 23.18 ± 0.1 | 27.89 ± 26. | 95.12 ± 0.6 | 62.02 ± 0.1 |
| ● Adam | 88.35 ± 5.7 | 27.77 ± 0.0 | 91.70 ± 0.3 | 83.32 ± 0.9 | 23.07 ± 0.1 | 54.92 ± 1.1 | 94.83 ± 0.3 | 61.97 ± 0.1 |
| ● LA(Mom.) | 87.16 ± 0.1 | 55.20 ± 0.9 | 88.56 ± 1.9 | 82.34 ± 1.0 | 25.38 ± 0.3 | 13.37 ± 15. | 95.67 ± 0.1 | 60.21 ± 0.3 |
| ● LA(RAdam) | 87.88 ± 5.8 | 34.28 ± 9.4 | 91.97 ± 0.2 | 82.92 ± 0.7 | 23.84 ± 0.2 | 55.84 ± 1.4 | **95.74 ± 0.2** | 61.72 ± 0.1 |
| ● Momentum | **87.03 ± 0.0** | 35.95 ± 11. | 90.15 ± 0.3 | 83.23 ± 0.5 | 24.33 ± 1.2 | 55.86 ± 1.3 | 95.55 ± 0.3 | 61.91 ± 0.1 |
| ● NAG | 87.08 ± 0.0 | 36.21 ± 11. | 90.09 ± 0.3 | 83.40 ± 0.6 | 23.86 ± 0.1 | **57.45 ± 0.6** | 95.55 ± 0.2 | 62.03 ± 0.1 |
| ● NAdam | 88.19 ± 5.7 | **27.72 ± 0.1** | 91.55 ± 0.4 | **83.91 ± 0.5** | **23.02 ± 0.1** | 54.30 ± 1.1 | 95.03 ± 0.3 | 61.98 ± 0.1 |
| ● RAdam | 88.18 ± 5.7 | 27.82 ± 0.1 | 91.66 ± 0.2 | 83.41 ± 0.6 | 23.12 ± 0.1 | 54.75 ± 1.0 | 94.92 ± 0.4 | 61.96 ± 0.1 |
| ● RMSProp | 87.40 ± 0.1 | 27.83 ± 0.1 | 88.84 ± 0.6 | 81.51 ± 0.5 | 23.18 ± 0.1 | 49.61 ± 1.4 | 94.41 ± 0.5 | **62.22 ± 0.1** |
| ● SGD | 87.98 ± 6.5 | 36.15 ± 10. | 91.69 ± 0.2 | 82.70 ± 1.3 | 23.94 ± 0.4 | 52.69 ± 1.1 | 95.11 ± 0.4 | 61.28 ± 0.1 |

APPENDIX REFERENCES

Laurence Aitchison. Bayesian filtering unifies adaptive and non-adaptive neural network optimization methods. In *Advances in Neural Information Processing Systems 33, NeurIPS*, 2020.

Rohan Anil, Vineet Gupta, Tomer Koren, Kevin Regan, and Yoram Singer. Second Order Optimization Made Practical. *arXiv preprint: 2002.09018*, 2020.

Imen Ayadi and Gabriel Turinici. Stochastic Runge-Kutta methods and adaptive SGD-G2 stochastic gradient descent. *arXiv preprint: 2002.09304*, 2020.

Kiwook Bae, Heechang Ryu, and Hayong Shin. Does Adam optimizer keep close to the optimal point?. *arXiv preprint: 1911.00289*, 2019.

Jiyang Bai and Jiawei Zhang. BGADAM: Boosting based Genetic-Evolutionary ADAM for Convolutional Neural Network Optimization. *arXiv preprint: 1908.08015*, 2019.

Lukas Balles and Philipp Hennig. Dissecting Adam: The Sign, Magnitude and Variance of Stochastic Gradients. In *35th International Conference on Machine Learning, ICML*, 2018.

Irwan Bello, Barret Zoph, Vijay Vasudevan, and Quoc V. Le. Neural Optimizer Search with Reinforcement Learning. In *34th International Conference on Machine Learning, ICML*, 2017.

Jeremy Bernstein, Yu-Xiang Wang, Kamyar Azizzadenesheli, and Animashree Anandkumar. SIGNSGD: Compressed Optimisation for Non-Convex Problems. In *35th International Conference on Machine Learning, ICML*, 2018.

Leonard Berrada, Andrew Zisserman, and M. Pawan Kumar. Training Neural Networks for and by Interpolation. In *37th International Conference on Machine Learning, ICML*, 2020.

Oleksandr Borysenko and Maksym Byshkin. CoolMomentum: A Method for Stochastic Optimization by Langevin Dynamics with Simulated Annealing. *arXiv preprint: 2005.14605*, 2020.

Aleksandar Botev, Hippolyt Ritter, and David Barber. Practical Gauss-Newton Optimisation for Deep Learning. In *34th International Conference on Machine Learning, ICML*, 2017.

Léon Bottou. Stochastic gradient descent tricks. In *Neural networks: Tricks of the trade*. Springer, 2012.

Chia-Yu Chen, Jungwook Choi, Daniel Brand, Ankur Agrawal, Wei Zhang, and Kailash Gopalakrishnan. AdaComp: Adaptive Residual Gradient Compression for Data-Parallel Distributed Training. In *32nd AAAI Conference on Artificial Intelligence, AAAI*, 2018.

Jinghui Chen, Dongruo Zhou, Yiqi Tang, Ziyan Yang, Yuan Cao, and Quanquan Gu. Closing the generalization gap of adaptive gradient methods in training deep neural networks. In *29th International Joint Conference on Artificial Intelligence, IJCAI*, 2020.

Xiangyi Chen, Sijia Liu, Ruoyu Sun, and Mingyi Hong. On the Convergence of A Class of Adam-Type Algorithms for Non-Convex Optimization. In *7th International Conference on Learning Representations, ICLR*, 2019a.

Yushu Chen, Hao Jing, Wenlai Zhao, Zhiqiang Liu, Ouyi Li, Liang Qiao, Wei Xue, Haohuan Fu, and Guangwen Yang. An Adaptive Remote Stochastic Gradient Method for Training Neural Networks. *arXiv preprint: 1905.01422*, 2019b.

Yushu Chen, Hao Jing, Wenlai Zhao, Zhiqiang Liu, Liang Qiao, Wei Xue, Haohuan Fu, and Guangwen Yang. NAMSG: An Efficient Method For Training Neural Networks. *arXiv preprint: 1905.01422*, 2019c.

Ziyi Chen and Yi Zhou. Momentum with Variance Reduction for Nonconvex Composition Optimization. *arXiv preprint: 2005.07755*, 2020.

Dami Choi, Christopher J. Shallue, Zachary Nado, Jaehoon Lee, Chris J. Maddison, and George E. Dahl. On Empirical Comparisons of Optimizers for Deep Learning. *arXiv preprint: 1910.05446*, 2019.

Aditya Devarakonda, Maxim Naumov, and Michael Garland. AdaBatch: Adaptive Batch Sizes for Training Deep Neural Networks. *arXiv preprint:* `1712.02029`, 2017.

Jianbang Ding, Xuancheng Ren, Ruixuan Luo, and Xu Sun. An Adaptive and Momental Bound Method for Stochastic Learning. *arXiv preprint:* `1910.12249`, 2019.

Timothy Dozat. Incorporating Nesterov Momentum into Adam. In *4th International Conference on Learning Representations, ICLR*, 2016.

Shiv Ram Dubey, Soumendu Chakraborty, Swalpa Kumar Roy, Snehasis Mukherjee, Satish Kumar Singh, and Bidyut Baran Chaudhuri. diffGrad: An Optimization Method for Convolutional Neural Networks. *IEEE Transactions on Neural Networks and Learning Systems*, 2020.

John Duchi, Elad Hazan, and Yoram Singer. Adaptive Subgradient Methods for Online Learning and Stochastic Optimization. *Journal of Machine Learning Research, JMLR*, 12, 2011.

Abraham J. Fetterman, Christina H. Kim, and Joshua Albrecht. SoftAdam: Unifying SGD and Adam for better stochastic gradient descent, 2019.

Boris Ginsburg, Patrice Castonguay, Oleksii Hrinchuk, Oleksii Kuchaiev, Vitaly Lavrukhin, Ryan Leary, Jason Li, Huyen Nguyen, and Jonathan M. Cohen. Stochastic Gradient Methods with Layer-wise Adaptive Moments for Training of Deep Networks. *arXiv preprint:* `1905.11286`, 2019.

Donald Goldfarb, Yi Ren, and Achraf Bahamou. Practical Quasi-Newton Methods for Training Deep Neural Networks. In *Advances in Neural Information Processing Systems 33, NeurIPS*, 2020.

Ian Goodfellow, Yoshua Bengio, and Aaron Courville. *Deep Learning*. MIT Press, 2016.

Priya Goyal, Piotr Dollár, Ross Girshick, Pieter Noordhuis, Lukasz Wesolowski, Aapo Kyrola, Andrew Tulloch, Yangqing Jia, and Kaiming He. Accurate, Large Minibatch SGD: Training ImageNet in 1 Hour. *arXiv preprint:* `1706.02677`, 2017.

Mikhail Grankin. RangerLars. `https://github.com/mgrankin/over9000`, 2020.

Vineet Gupta, Tomer Koren, and Yoram Singer. Shampoo: Preconditioned Stochastic Tensor Optimization. In *35th International Conference on Machine Learning, ICML*, 2018.

Kaiming He, Xiangyu Zhang, Shaoqing Ren, and Jian Sun. Deep Residual Learning for Image Recognition. In *IEEE Computer Society Conference on Computer Vision and Pattern Recognition*, 2016.

João F. Henriques, Sébastien Ehrhardt, Samuel Albanie, and Andrea Vedaldi. Small Steps and Giant Leaps: Minimal Newton Solvers for Deep Learning. In *IEEE/CVF International Conference on Computer Vision, ICCV*, 2019.

Byeongho Heo, Sanghyuk Chun, Seong Joon Oh, Dongyoon Han, Sangdoo Yun, Youngjung Uh, and Jung-Woo Ha. Slowing Down the Weight Norm Increase in Momentum-based Optimizers. *arXiv preprint:* `2006.08217`, 2020.

Jeremy Howard and Sebastian Ruder. Universal Language Model Fine-tuning for Text Classification. In *56th Annual Meeting of the Association for Computational Linguistics*, 2018.

Yifan Hu, Siqi Zhang, Xin Chen, and Niao He. Biased Stochastic First-Order Methods for Conditional Stochastic Optimization and Applications in Meta Learning. In *Advances in Neural Information Processing Systems 33, NeurIPS*, 2020.

Yuzheng Hu, Licong Lin, and Shange Tang. Second-order Information in First-order Optimization Methods. *arXiv preprint:* `1912.09926`, 2019.

Haiwen Huang, Chang Wang, and Bin Dong. Nostalgic Adam: Weighting More of the Past Gradients When Designing the Adaptive Learning Rate. In *28th International Joint Conference on Artificial Intelligence, IJCAI*, 2019.

Xunpeng Huang, Hao Zhou, Runxin Xu, Zhe Wang, and Lei Li. Adaptive Gradient Methods Can Be Provably Faster than SGD after Finite Epochs. *arXiv preprint: 2006.07037*, 2020.

Yasutoshi Ida, Yasuhiro Fujiwara, and Sotetsu Iwamura. Adaptive Learning Rate via Covariance Matrix Based Preconditioning for Deep Neural Networks. In *26th International Joint Conference on Artificial Intelligence, IJCAI*, 2017.

Wendyam Eric Lionel Ilboudo, Taisuke Kobayashi, and Kenji Sugimoto. TAdam: A Robust Stochastic Gradient Optimizer. *arXiv preprint: 2003.00179*, 2020.

Zhanhong Jiang, Aditya Balu, Sin Yong Tan, Young M Lee, Chinmay Hegde, and Soumik Sarkar. On Higher-order Moments in Adam. *arXiv preprint: 1910.06878*, 2019.

Tyler B. Johnson, Pulkit Agrawal, Haijie Gu, and Carlos Guestrin. AdaScale SGD: A User-Friendly Algorithm for Distributed Training, 2020.

Dominic Kafka and Daniel Wilke. Gradient-only line searches: An Alternative to Probabilistic Line Searches. *arXiv preprint: 1903.09383*, 2019.

Chad Kelterborn, Marcin Mazur, and Bogdan V. Petrenko. Gravilon: Applications of a New Gradient Descent Method to Machine Learning. *arXiv preprint: 2008.11370*, 2020.

Nitish Shirish Keskar and Richard Socher. Improving Generalization Performance by Switching from Adam to SGD. *arXiv preprint: 1712.07628*, 2017.

Mohammad Emtiyaz Khan, Didrik Nielsen, Voot Tangkaratt, Wu Lin, Yarin Gal, and Akash Srivastava. Fast and Scalable Bayesian Deep Learning by Weight-Perturbation in Adam. In *35th International Conference on Machine Learning, ICML*, 2018.

Diederik P. Kingma and Jimmy Ba. Adam: A Method for Stochastic Optimization. In *3rd International Conference on Learning Representations, ICLR*, 2015.

Kfir Yehuda Levy, Alp Yurtsever, and Volkan Cevher. Online Adaptive Methods, Universality and Acceleration. In *Advances in Neural Information Processing Systems 31, NeurIPS*, 2018.

Wenjie Li, Zhaoyang Zhang, Xinjiang Wang, and Ping Luo. AdaX: Adaptive Gradient Descent with Exponential Long Term Memory. *arXiv preprint: 2004.09740*, 2020a.

Zhize Li, Hongyan Bao, Xiangliang Zhang, and Peter Richtárik. PAGE: A Simple and Optimal Probabilistic Gradient Estimator for Nonconvex Optimization. *arXiv preprint: 2008.10898*, 2020b.

Liang Liu and Xiaopeng Luo. A New Accelerated Stochastic Gradient Method with Momentum. *arXiv preprint: 2006.00423*, 2020.

Liyuan Liu, Haoming Jiang, Pengcheng He, Weizhu Chen, Xiaodong Liu, Jianfeng Gao, and Jiawei Han. On the variance of the adaptive learning rate and beyond. In *8th International Conference on Learning Representations, ICLR*, 2020.

Pietro Longhi. Wall crossing invariants from spectral networks. *Annales Henri Poincaré*, 19(3), 2017.

Ilya Loshchilov and Frank Hutter. SGDR: Stochastic Gradient Descent with Warm Restarts. In *5th International Conference on Learning Representations, ICLR*, 2017.

Ilya Loshchilov and Frank Hutter. Decoupled weight decay regularization. In *7th International Conference on Learning Representations, ICLR*, 2019.

Liangchen Luo, Yuanhao Xiong, Yan Liu, and Xu Sun. Adaptive Gradient Methods with Dynamic Bound of Learning Rate. In *7th International Conference on Learning Representations, ICLR*, 2019.

Jerry Ma and Denis Yarats. Quasi-hyperbolic momentum and Adam for deep learning. In *7th International Conference on Learning Representations, ICLR*, 2019.

Maren Mahsereci. *Probabilistic Approaches to Stochastic Optimization*. Ph.D. Thesis, University of Tuebingen, 2018.

Itzik Malkiel and Lior Wolf. MTAdam: Automatic Balancing of Multiple Training Loss Terms. *arXiv preprint: 2006.14683*, 2020.

James Martens and Roger Grosse. Optimizing Neural Networks with Kronecker-Factored Approximate Curvature. In *32nd International Conference on Machine Learning, ICML*, 2015.

Mahesh Chandra Mukkamala and Matthias Hein. Variants of RMSProp and Adagrad with Logarithmic Regret Bounds. In *34th International Conference on Machine Learning, ICML*, 2017.

Maximus Mutschler and Andreas Zell. Parabolic Approximation Line Search for DNNs. In *Advances in Neural Information Processing Systems 33, NeurIPS*, 2020.

Parvin Nazari, Davoud Ataee Tarzanagh, and George Michailidis. DADAM: A Consensus-based Distributed Adaptive Gradient Method for Online Optimization. *arXiv preprint: 1901.09109*, 2019.

Yurii Nesterov. A method for solving the convex programming problem with convergence rate O(1/k^2). *Soviet Mathematics Doklady*, 27, 1983.

Francesco Orabona and Dávid Pál. Scale-Free Algorithms for Online Linear Optimization. In *Algorithmic Learning Theory - 26th International Conference, ALT*, 2015.

Antonio Orvieto, Jonas Köhler, and Aurélien Lucchi. The Role of Memory in Stochastic Optimization. In *35th Conference on Uncertainty in Artificial Intelligence, UAI*, 2019.

B. T. Polyak. Some methods of speeding up the convergence of iteration methods. *USSR Computational Mathematics and Mathematical Physics*, 4(5), 1964.

Konpat Preechakul and Boonserm Kijsirikul. CProp: Adaptive Learning Rate Scaling from Past Gradient Conformity. *arXiv preprint: 1912.11493*, 2019.

Sashank J. Reddi, Satyen Kale, and Sanjiv Kumar. On the Convergence of Adam and Beyond. In *6th International Conference on Learning Representations, ICLR*, 2018.

Herbert Robbins and Sutton Monro. A Stochastic Approximation Method. *The Annals of Mathematical Statistics*, 22(3), 1951.

Michal Rolínek and Georg Martius. L4: Practical loss-based stepsize adaptation for deep learning. In *Advances in Neural Information Processing Systems 31, NeurIPS*, 2018.

Arnold Salas, Samuel Kessler, Stefan Zohren, and Stephen Roberts. Practical Bayesian Learning of Neural Networks via Adaptive Subgradient Methods. *arXiv preprint: 1811.03679*, 2018.

Pedro Savarese, David McAllester, Sudarshan Babu, and Michael Maire. Domain-independent Dominance of Adaptive Methods. *arXiv preprint: 1912.01823*, 2019.

Tom Schaul and Yann LeCun. Adaptive learning rates and parallelization for stochastic, sparse, non-smooth gradients. In *1st International Conference on Learning Representations, ICLR*, 2013.

Tom Schaul, Sixin Zhang, and Yann LeCun. No more pesky learning rates. In *30th International Conference on Machine Learning, ICML*, 2013.

Fanhua Shang, Kaiwen Zhou, Hongying Liu, James Cheng, Ivor W. Tsang, Lijun Zhang, Dacheng Tao, and Licheng Jiao. VR-SGD: A Simple Stochastic Variance Reduction Method for Machine Learning. *IEEE Trans. Knowl. Data Eng.*, 32(1), 2020.

Leslie N. Smith. Cyclical Learning Rates for Training Neural Networks. In *IEEE Winter Conference on Applications of Computer Vision, WACV*, 2017.

Leslie N. Smith and Nicholay Topin. Super-Convergence: Very Fast Training of Neural Networks Using Large Learning Rates. *arXiv preprint: 1708.07120*, 2017.

Huikang Sun, Lize Gu, and Bin Sun. Adathm: Adaptive Gradient Method Based on Estimates of Third-Order Moments. In *4th IEEE International Conference on Data Science in Cyberspace, DSC*, 2019.

Wonyong Sung, Iksoo Choi, Jinhwan Park, Seokhyun Choi, and Sungho Shin. S-SGD: Symmetrical Stochastic Gradient Descent with Weight Noise Injection for Reaching Flat Minima. *arXiv preprint: 2009.02479*, 2020.

Conghui Tan, Shiqian Ma, Yu-Hong Dai, and Yuqiu Qian. Barzilai-Borwein Step Size for Stochastic Gradient Descent. In *Advances in Neural Information Processing Systems 29, NIPS*, 2016.

Zeyi Tao, Qi Xia, and Qun Li. A new perspective in understanding of Adam-Type algorithms and beyond, 2019.

Brian Teixeira, Birgi Tamersoy, Vivek Singh, and Ankur Kapoor. Adaloss: Adaptive Loss Function for Landmark Localization. *arXiv preprint: 1908.01070*, 2019.

Tijmen Tieleman and Geoffrey Hinton. Lecture 6.5—RMSProp: Divide the gradient by a running average of its recent magnitude, 2012.

Qianqian Tong, Guannan Liang, and Jinbo Bi. Calibrating the Adaptive Learning Rate to Improve Convergence of ADAM. *arXiv preprint: 1908.00700*, 2019.

Phuong Thi Tran and Le Trieu Phong. On the Convergence Proof of AMSGrad and a New Version. *IEEE Access*, 7, 2019.

Rasul Tutunov, Minne Li, Alexander I. Cowen-Rivers, Jun Wang, and Haitham Bou-Ammar. Compositional ADAM: An Adaptive Compositional Solver. *arXiv preprint: 2002.03755*, 2020.

Ashish Vaswani, Noam Shazeer, Niki Parmar, Jakob Uszkoreit, Llion Jones, Aidan N. Gomez, Łukasz Kaiser, and Illia Polosukhin. Attention Is All You Need. In *Advances in Neural Information Processing Systems 30, NIPS*, 2017.

Sharan Vaswani, Aaron Mishkin, Issam H. Laradji, Mark Schmidt, Gauthier Gidel, and Simon Lacoste-Julien. Painless Stochastic Gradient: Interpolation, Line-Search, and Convergence Rates. In *Advances in Neural Information Processing Systems 32, NeurIPS*, 2019.

Thijs Vogels, Sai Praneeth Karimireddy, and Martin Jaggi. PowerSGD: Practical Low-Rank Gradient Compression for Distributed Optimization. In *Advances in Neural Information Processing Systems 32, NeurIPS*, 2019.

Bao Wang, Tan M. Nguyen, Andrea L. Bertozzi, Richard G. Baraniuk, and Stanley J. Osher. Scheduled restart momentum for accelerated stochastic gradient descent. *arXiv preprint: 2002.10583*, 2020a.

Dong Wang, Yicheng Liu, Wenwo Tang, Fanhua Shang, Hongying Liu, Qigong Sun, and Licheng Jiao. signADAM++: Learning Confidences for Deep Neural Networks. In *International Conference on Data Mining Workshops, ICDM*, 2019a.

Guanghui Wang, Shiyin Lu, Quan Cheng, Weiwei Tu, and Lijun Zhang. SAdam: A Variant of Adam for Strongly Convex Functions. In *8th International Conference on Learning Representations, ICLR*, 2020b.

Jiaxuan Wang and Jenna Wiens. AdaSGD: Bridging the gap between SGD and Adam. *arXiv preprint: 2006.16541*, 2020.

Shipeng Wang, Jian Sun, and Zongben Xu. HyperAdam: A Learnable Task-Adaptive Adam for Network Training. In *33rd AAAI Conference on Artificial Intelligence, AAAI*, 2019b.

Less Wright. Deep Memory. https://github.com/lessw2020/Best-Deep-Learning-Optimizers/tree/master/DeepMemory, 2020a.

Less Wright. Ranger. https://github.com/lessw2020/Ranger-Deep-Learning-Optimizer, 2020b.

Xiaoxia Wu, Rachel Ward, and Léon Bottou. WNGrad: Learn the Learning Rate in Gradient Descent. *arXiv preprint:* `1803.02865`, 2018.

Cong Xie, Oluwasanmi Koyejo, Indranil Gupta, and Haibin Lin. Local AdaAlter: Communication-Efficient Stochastic Gradient Descent with Adaptive Learning Rates. *arXiv preprint:* `1911.09030`, 2019.

Chen Xing, Devansh Arpit, Christos Tsirigotis, and Yoshua Bengio. A Walk with SGD. *arXiv preprint:* `1802.08770`, 2018.

Yangyang Xu. Momentum-based variance-reduced proximal stochastic gradient method for composite nonconvex stochastic optimization. *arXiv preprint:* `2006.00425`, 2020.

Minghan Yang, Dong Xu, Yongfeng Li, Zaiwen Wen, and Mengyun Chen. Structured Stochastic Quasi-Newton Methods for Large-Scale Optimization Problems. *arXiv preprint:* `2006.09606`, 2020.

Zhewei Yao, Amir Gholami, Sheng Shen, Kurt Keutzer, and Michael W. Mahoney. ADAHESSIAN: An Adaptive Second Order Optimizer for Machine Learning. *arXiv preprint:* `2006.00719`, 2020.

Yang You, Igor Gitman, and Boris Ginsburg. Large Batch Training of Convolutional Networks. *arXiv preprint:* `1708.03888`, 2017.

Yang You, Jing Li, Sashank Reddi, Jonathan Hseu, Sanjiv Kumar, Srinadh Bhojanapalli, Xiaodan Song, James Demmel, Kurt Keutzer, and Cho-Jui Hsieh. Large Batch Optimization for Deep Learning: Training BERT in 76 minutes. In *8th International Conference on Learning Representations, ICLR*, 2020.

Jihun Yun, Aurelie C. Lozano, and Eunho Yang. Stochastic Gradient Methods with Block Diagonal Matrix Adaptation. *arXiv preprint:* `1905.10757`, 2019.

Manzil Zaheer, Sashank J. Reddi, Devendra Singh Sachan, Satyen Kale, and Sanjiv Kumar. Adaptive Methods for Nonconvex Optimization. In *Advances in Neural Information Processing Systems 31, NeurIPS*, 2018.

Matthew D. Zeiler. ADADELTA: An Adaptive Learning Rate Method. *arXiv preprint:* `1212.5701`, 2012.

Guodong Zhang, Shengyang Sun, David Duvenaud, and Roger Grosse. Noisy Natural Gradient as Variational Inference. In *35th International Conference on Machine Learning, ICML*, 2018.

Jian Zhang and Ioannis Mitliagkas. YellowFin and the Art of Momentum Tuning. In *Machine Learning and Systems, MLSys*, 2019.

Jiawei Zhang and Fisher B. Gouza. GADAM: Genetic-Evolutionary ADAM for Deep Neural Network Optimization. *arXiv preprint:* `1805.07500`, 2018.

Jingzhao Zhang, Sai Praneeth Karimireddy, Andreas Veit, Seungyeon Kim, Sashank J Reddi, Sanjiv Kumar, and Suvrit Sra. Why are adaptive methods good for attention models? In *Advances in Neural Information Processing Systems 33, NeurIPS*, 2020.

Michael R. Zhang, James Lucas, Geoffrey Hinton, and Jimmy Ba. Lookahead Optimizer: k steps forward, 1 step back. *Advances in Neural Information Processing Systems 32, NeurIPS*, 2019.

Zijun Zhang, Lin Ma, Zongpeng Li, and Chuan Wu. Normalized Direction-preserving Adam. *arXiv preprint:* `1709.04546`, 2017.

Shen-Yi Zhao, Yin-Peng Xie, and Wu-Jun Li. Stochastic Normalized Gradient Descent with Momentum for Large Batch Training. *arXiv preprint:* `2007.13985`, 2020.

Bingxin Zhou, Xuebin Zheng, and Junbin Gao. ADAMT: A Stochastic Optimization with Trend Correction Scheme. *arXiv preprint:* `2001.06130`, 2020.

Zhiming Zhou, Qingru Zhang, Guansong Lu, Hongwei Wang, Weinan Zhang, and Yong Yu. AdaShift: Decorrelation and Convergence of Adaptive Learning Rate Methods. In *7th International Conference on Learning Representations, ICLR*, 2019.

Liu Ziyin, Zhikang T. Wang, and Masahito Ueda. LaProp: a Better Way to Combine Momentum with Adaptive Gradient. *arXiv preprint: 2002.04839*, 2020.

