# OpenReview forum: "Descending through a Crowded Valley — Benchmarking Deep Learning Optimizers"
_ICLR.cc/2021/Conference — Reject_

### Official Review · AnonReviewer4 · 2020-10-26
**Very well done with one experimental concern**

**Rating:** 9
**Confidence:** 5

**Review:**

Overview:
Overall I believe this paper is extremely well written and organized. The introduction and limitations are very useful groundings of optimization research, and I hope that the community reads this paper and internalizes its message of making more meaningful research in optimization (instead of yet-another-Adam-variant!).

In a previous comment I brought up a serious concern about the tuning ranges of momentum-like parameters, which I believe could bias the results towards optimizers that were tuned with ranges whose lower end was 0.5 (Adam, AMSBound, AMSGrad, AdaBound, LA(RAdam), NAdam, RAdam). Besides this concern, I overall would normally argue for a very strong acceptance, but until this is corrected I am unsure I can recommend accepting. I look forward to discussing correcting this with the authors however, and am willing to dramatically raise my score!

**NOTE: updated score after seeing author replies and updated draft. I believe that this work is exemplary in terms of being careful about baseline construction, something that is unfortunately too often overlooked in our field. Additionally, it rigorously highlights another important point that I believe many often overlook, that "there are now enough optimizers"; community effort should be diverted from introducing small variations around Adam and instead invest focus on more meaningful improvements in scaling machine learning optimization. I do still believe that ImageNet and a larger transformer experiment would be extremely valuable to add to a later version, and hope the authors can eventually secure the computing power to add these.**

Pros:
-Table 2 is extremely useful and I think should be additionally put on a GitHub where it can be updated with links to papers.
-The authors clearly put a lot of careful thought into how to study and present these results, taking into account numerous caveats that almost all other papers totally ignore, such as the limitations of their study, the importance of tuning ranges, learning rate schedules, etc.

Concerns:
-I am very understanding and sympathetic to the issue of compute constraints, but I do believe that the study would be even more useful if a ResNet-50/Imagenet and a Transformer pipeline were used. The authors discuss that GANs and RL are not included, and those seem like very different types of optimization to me so I am more understanding of not including them. If the authors are academics then I have seen researchers have success in the past with getting grants from cloud providers, namely the lottery ticket hypothesis line of work and the TF Research Cloud (I found out about it through this paper https://openreview.net/forum?id=S1gSj0NKvB), which can easily be used to heavily tune ImageNet and Transformer runs. Outside of that, according to this benchmark https://dawn.cs.stanford.edu/benchmark/ImageNet/train.html, the cost of ImageNet runs has come down considerably in recent years, to $10-20 per run.
-The experimental results could partially be explained by the No Free Lunch theorem, and the authors could at least reference this in the section.
-What regularization, if any, was used for these problems, and was that also tuned? One could argue that, while optimization and regularization are in theory orthogonal to each other in what they try to accomplish (train vs test performance), they are both part of the update rule whose hyperparameters are being tuned. Also I believe it is important to be careful and note if coupled or decoupled weight decay/L2 is being used is important; I assume it is coupled because the DeepOBS code that is referenced uses L2 regularization, which means that optimizers that use preconditioning (Adam-like optimizers) could be impacted by this differently than those that do not (SGD/Momentum).

Writing:
-In the “Tuning method” paragraph in section 2.3, “In case there is no prior knowledge provided in the op.cit. we chose” seems like a syntax error.

Prior work:
The authors do a very thorough literature search, and properly reference and discuss similar prior studies.

Additional feedback, comments, suggestions for improvement and questions for the authors:
-Awesome job providing per-step values for results, it would be further useful to have code that could easily plot them side-by-side so that future researchers would be further encouraged to include them in their figures.
-May be worth noting that, in addition to the optimizer hyperparameters, one could also tune the batch normalization momentum/epsilon, for additional performance gains.
-Figure 3 seems very useful, however I believe it would be much better presented as a series of box plots. A nice recent example of this is Figure 2 in https://arxiv.org/abs/1906.02530. This could also be done for Figure 4 where each nested box is an individual optimizer, and it shows the distribution over runs for each optimizer for each problem.
-The trapez schedule always seems to be the best, and I wonder if this is due to only one of the learning rate ramp up, which has been shown to be beneficial to stabilize training (although it is unclear if this is required), or the learning rate becoming quite small at the end, which has been shown to be necessary so that the optimizer can better learn the noisy directions of the objective. It would be useful for future work to consider each of these learning rate schedules (ramp-up, ramp-down) separately, although that requires even more compute.

---

> ### Author Response · Authors · 2020-11-12
> **Response to the Comments of Reviewer 4**
>
> We want to thank you for the very constructive feedback.
>
> As we already wrote below, we agree with your point about the momentum tuning ranges. This is an absolutely valid concern that we will address. We are currently trying to get our hands on as much hardware as possible to update the corresponding results. From our current estimate, it will be a nail-biter whether we can finish every single re-run by the end of this feedback session.
> Thus, it would be very valuable to us if you could give us direct feedback on whether our current plans will seem satisfactory to you, or whether you want us to do things differently. Here is what we plan to do:
>
> - We will change the tuning distribution of Adadelta, Lookahead Momentum, Momentum, NAG, and RMSProp. Instead of tuning $\rho$ with $LU(1e-3, 1)$, we will now tune $1-\rho$ with $LU(1e-4,1)$ as you suggested.
> - We will start with all runs affecting the results in Figures 2 and 4 in the main text.
> - After that, we will iteratively update all remaining Figures and the Appendix when the corresponding results arrive.
>
> We will post updates on OpenReview as soon as they arrive and can show the update figures in our anonymous git repo or in the updated pdf.
>
> Thanks again for your willingness to debate with us. We’re investing significant resources in trying to address your concerns and want to make sure this effort is worthwhile.
>
> We also greatly appreciate the more minor suggestions. We have commented above, in the reply to Reviewer 1, on the use of larger data sets, but putting Table 2 on Github with links, mentioning the No Free Lunch Theorem, changing "op.cit." to something more comprehensible, and commenting on the used regularization are great suggestions that we will implement immediately.

---

> > ### Comment · AnonReviewer4 · 2020-11-13
> > **Tuning ranges**
> >
> > I believe that investing would be worthwhile!
> >
> > One nit about the tuning ranges, for (Adam, AMSBound, AMSGrad, AdaBound, LA(RAdam), NAdam, RAdam) the momentum-like parameter is tuned on LU(0.5, 0.999), which would be the equivalent of tuning $1 - \rho$ on LU(1e-4, 0.5), but you're tuning on LU(1e-4, 1.0). In practice this likely won't matter, but I would highlight it in the text.

---

### Official Review · AnonReviewer3 · 2020-10-27
**Review of "Descending through a Crowded Valley — Benchmarking Deep Learning Optimizers"**

**Rating:** 4
**Confidence:** 5

**Review:**

Summary:
This paper benchmarks popular optimizers for training neural networks. The experiments consider all possible combinations of 3 different tuning budgets, and 4 different fixed learning rate schedules on 8 deep learning workloads for 14 optimizers. The paper highlights two main observations: 1) there is no clear dominating optimizer, and 2) selecting from a pool of optimizers with their default parameters is often as good as tuning a fixed optimizer.
The main contribution of this work is the open-sourced experiment results for a multitude of cases which can serve as a baseline for future research in optimizers for deep learning.

Strengths:
- The biggest strength of this work is that all the details regarding the benchmarking protocol are presented and justified. In addition to this, the caveats of the protocol are stated explicitly. The explicit and transparent nature of this work can help practitioners make more informed decisions, and prevent them from misinterpreting the results to something more than what is presented.
- To my knowledge, this is the first paper to compare a large number of optimizers, selected based on popularity in the research community.
- The problems considered are of varying difficulty, and includes tasks other than image classification.
- Different levels of tuning budgets are considered, with the smallest budget being 1 trial (evaluating the default setting), and the largest being 50 trials.

Weaknesses
- The current tuning procedure is unstable. Tuning with the seed fixed is like optimizing for the specific seed. Furthermore, the hyperparameters that produce the best performance for a specific seed tends to be more unstable (which the authors agree to in appendix C); evaluating such an unstable setting on different seeds unnecessarily penalizes the optimizer. What makes more sense to me is to tune with the same number of trials, with each trial having a different seed, and using bootstrapping to compute the statistics (mean, standard deviation, etc). What we want to see with the tuning experiments is how well the optimizer can do (as an upper bound), and how stable it is. With the current approach, it’s hard to observe the best performance, at which point, I’m not sure how meaningful the error measurements are. All in all, I think it’s more meaningful to show how varying the best optimizer performance can be when tuning with a different set of seeds (since everyone uses different seeds), than to show the variance of a specific set of hyperparameters that is most likely unstable, on many seeds.
- I believe the experiments lack results for the “well-tuned” case. The optimizers all use a fixed hyperparameter search range for all problems, which can’t be competitive over different tasks of varying difficulty. I understand that this study assumes the model practitioner to be someone who doesn’t have prior knowledge about the optimizer, let alone the search ranges. However, I think it’s reasonable to believe that a practitioner would try to verify the search range by testing some hyperparameter values before committing 25 or 50 trials to the search range. Likewise, it would be useful to see results with a more calibrated search space per test problem. This can be done with the 50 trial budget by, for example, using 25 on a wide search space, and the other 25 on a more refined search space. At the very least, it would be useful to see the performance vs hyperparameter value plotted for the existing experiments to see whether the ranges could have been trivially improved (for example, if the performance tends to increase/decrease with the learning rate, but the best performance lied on the boundary of the search space, the range could have been shifted). This sort of tuning procedure is not unknown in the community. See [1, 2, 3].

Currently, I think the weaknesses outweigh the strengths of the paper. It is my understanding that optimizer comparisons should be done between reasonably good versions of the optimizers, and I think better versions of the optimizers could have been presented with a different methodology, given the same computational budget.



[1] Wilson, Ashia C., et al. "The marginal value of adaptive gradient methods in machine learning." Advances in neural information processing systems. 2017.

[2] Shallue, Christopher J., et al. "Measuring the effects of data parallelism on neural network training." arXiv preprint arXiv:1811.03600 (2018).

[3] Choi, Dami, et al. "On empirical comparisons of optimizers for deep learning." arXiv preprint arXiv:1910.05446 (2019).


Update:

I have read over the changes made by the authors, and also the other reviewer’s responses. I am maintaining my score, because I don’t think the current version of the paper is enough of a contribution to get accepted. As mentioned in my responses below, I would be happy to accept a future version of the paper that addresses my comments above.

---

> ### Author Response · Authors · 2020-11-12
> **Response to the Comments of Reviewer 3**
>
> Dear Reviewer 3,
>
> we want to thank you for your time reviewing our paper and providing specific feedback.
>
> It is great to hear that you found our benchmark well presented and justified in all its details. We also took great care to highlight its limitations and we are thus happy to read that you listed this as a strength of this work.
>
> Regarding the stability of our tuning procedure, we tried to model our process on an average practitioner. We expect them to do some kind of hyperparameter search (random search in our case) using a single run per setting (i.e. using a single seed). Once they found the best-working configuration, however, they might train the network multiple times, possibly using operations that would effectively change the random seed (e.g. using different data sampling, slightly changing the architecture, etc.). Our benchmark also offers the ability to assess how stable the optimizer would behave in this setting.
> Note, however, that it is still possible to assess "how well the optimizer can do (as an upper bound)" using just the seed that has been used for tuning. For example, Figure 11 shows the results of only the tuning seed and everyone is welcome to use our open-source data to delve deeper.
>
> When designing our benchmark we took care to not only care about performance but also about ease-of-use of the optimizer. If one can re-use the same search space for different problems, it makes this optimizer obviously easier to use. One can just run the hyperparameter search on the weekend, and get the results by Monday irrespective of the problem. This should be reflected in the benchmark, at least somehow.
> We did consider other hyperparameter tuning methods, such as the "two-stage random search" you are proposing or Bayesian methods. However, we decided against these more elaborate schemes mainly because they require additional human decisions and can thus introduce bias.
> Note, that, for example, the paper by Choi et al. (2019) has been criticized for its use of problem-dependent search spaces on OpenReview and by Sivaprasad et al. (2020). This is not to say that our method is superior, but that each method has its pros and cons.
>
> We want to also invite you to engage in the discussion we have below with Reviewer 4, in which we are trying to find a constructive way to address as many requests made by you and the other reviewers as possible within the rebuttal phase.

---

> > ### Comment · AnonReviewer4 · 2020-11-15
> > **Search space selection procedure**
> >
> > I have seen that many times in practice, machine learning users tune over a fixed set of hyperparameter points independent of dataset, and only sometimes increase the ranges. However, for problems where significant investment must be made, search spaces are refined through trial and error, at the high cost of compute. So I agree with Reviewer 3 that it is valid and not unheard of to do a search space refinement, however I also agree with the authors here in using a fixed search space for all problems. If using problem-dependent or hand-refined search spaces, it becomes difficult to tease apart where the performance boosts are coming from: the optimizer or the search space selection process? For example, in your 25 exploratory then 25 refinement trial scenario, would one see significant improvements in optimizer performance if 10 trials were used to explore and then 40 to refine (or vice versa)?
> >
> > In general, I believe that as long as authors are upfront about their search space selection process, one can argue for either approach, but perhaps the refinement approach would best be studied in a separate work on search space selection. In the current draft, the authors are explicit in their discussion of these nuances and highlight the possible drawbacks of their choice, so I do not believe they should be penalized for this.

---

> > > ### Comment · AnonReviewer3 · 2020-11-24
> > > **It's impossible to tease apart the optimizer and search space selection process**
> > >
> > > It’s impossible to talk about empirical performance of optimizers without discussing the search space, because optimizer performance is very heavily dependent on the search space. Therefore, it doesn't make sense to tease apart the optimizer and the search space selection process. This is exactly why comparing optimizers empirically is so difficult-- there seems to be no real answer as to what is fair. But this doesn’t mean that we should be complacent about choosing suboptimal ranges. We know that a given optimizer’s optimal hyperparameter values change based on the architecture and the dataset. I think a simple two-step procedure that is consistent among all optimizers is good enough for me, and is already so much better than sticking to a fixed range. As I mentioned in my response to the authors, this method doesn’t introduce any more bias than what already exists with the chosen fixed ranges. It’s just not acceptable for there to be a possibility where different conclusions could have been made with a slightly more involved setup that doesn’t take any more computation.

---

> > ### Comment · AnonReviewer3 · 2020-11-24
> > **I'm not convinced by the response**
> >
> > I appreciate the authors’ effort to account for different use cases of optimizers, and to model the “average practitioner”. However, following the experimental protocol of a fictitious group of people is not only imprecise, but also propagates and justifies bad practice, like tuning over a single seed, or using the same search ranges across all problems. I think this paper would be much more useful if it provided practical guidelines that ultimately result in better empirical results, rather than follow the authors’ idea of a model practitioner.
> >
> > Going back to my review, I think it’s clear that doing bootstrapping to estimate the metrics, will give better results than tuning on a single seed and evaluating on a different set of seeds. Not only this, but tuning with varying seeds doesn’t cost more computationally, and the resulting measurements are robust to the choice of seed that any given person would choose. The existence of Figure 11 doesn’t satisfy me, since the main body still contains results using the original tuning strategy. Furthermore, the mean of maxes (over different seeds) and the error measurements aren’t addressed by Figure 11, which includes only the max over a single seed.
> >
> > On a similar note, using a more tailored search space per problem should yield better performance than using a fixed global search space. The topic of hyperparameter search spaces is tricky but important to discuss, because optimizer performance is heavily dependent on the search space. Regarding the authors’ response, I think it’s impossible to take out the human factor in deciding the search spaces, unless we have an infinite amount of resources. In fact, the search spaces used in the paper were arbitrarily set by the authors, or chosen by a group of people through many empirical studies. Is the worry that by a human further refining the search ranges, that they will unknowingly bias certain methods over others? Isn’t this already the case, with more known optimizers getting a better range, since they were used many times in the past and therefore, has a more refined search space, compared to the more recent ones whose search ranges haven’t been explored? I feel like refining the search space per problem is a way to address this implicit bias, and make the comparisons more fair. I agree with the authors that no one method is perfect, and they each have pros and cons. However, I would opt for a method that yields better results, and practically useful information, like the specific search ranges that worked well for a given optimizer and dataset pair. The “two-stage” method that I proposed in my review, doesn’t add any more computation, and is simple compared to bayesian optimization.
> >
> > In summary, this paper simulates the “careful practitioner who does not have … a broad range of personal experiences,” which resulted in unnecessary penalization of optimizers, which could have been avoided with no additional computational effort. I’m not convinced by the author’s response, and additionally, agree with other reviewers’ concern about the limitation of the chosen workloads. Therefore, I am lowering my score, and recommend rejection.

---

> > > ### Author Response · Authors · 2020-11-24
> > > **Reply**
> > >
> > > We are sorry to hear that our response does not seem to have satisfied your concerns.
> > > If it helps, we are of course happy to move Figure 11 from the appendix to the main paper.
> > >
> > > However, your main concern seems to be our use of universal search spaces instead of adapting them individually. You agree that “no one method is perfect, and they each have pros and cons” and we are open about possible drawbacks of our (unavoidable) choices. We could state them more extensively in the paper if this would address your concerns.
> > >
> > > Allow us to defend our tuning method with one example. Let’s assume two optimizers A and B which both result in the same performance if tuned well. Optimizer A can always be tuned in the interval [1e-3,1] and Optimizer B’s interval depends on the problem, it might be [1e-6,1e-3] or [1e-1,1e2]. Wouldn’t you agree that Optimizer A is easier to use? If we use 25 runs to find the interval and 25 runs to tune it, both optimizers would look the same (given our admittedly simplified assumptions).
> > > Again, this is not to say that our method is the only valid choice! But yours isn’t either: In the same way that you suggested a 25/25 split (of tuning, adjusting the search space, and tuning again), another reviewer might prefer a 40/10 (better coverage) or a 10/40 (better tuning) split. Or a 12/13 split for the smaller budget. Or criticize this method as increasing the variance of the results and introducing yet another arbitrary choice. For example, the paper by Choi et al. (2019) that you reference, uses the problem-dependent search space you suggested, and their choice has also been debated.
> > >
> > > We accept that any benchmark will always invite criticism. But if the bar is to satisfy everyone, then these papers become virtually impossible. We agree with Reviewer 4 that perhaps the approach you suggest “would best be studied in a separate work on search space selection” that could provide another data point for benchmark optimizers additional to our work.

---

> > > > ### Comment · AnonReviewer3 · 2020-11-24
> > > > **The paper has a narrow scope and does not address some of the concerns that the authors claim**
> > > >
> > > > All papers involve making choices, which comes with drawbacks and caveats. It’s also the job of the reviewers to judge whether the choices are justified and adequate enough to be published. In my opinion, the choices made in this paper are not adequate and justified enough.
> > > >
> > > > Currently, the paper presents itself with a very narrow and limited scope. The experiments are only applicable to those who tune over a single seed, those who only have enough resources to use default parameter values, and those who tune over the arbitrary fixed search range decided by the authors and some other non-referenced group of people. I personally don’t think a paper that targets a very specific group of people, who even the authors think are inexperienced, is enough to get accepted to a conference.
> > > >
> > > > The authors seem to claim that their experiments are more than what they are; for instance, in their earlier reply, they mentioned that they “took care to not only care about performance but also ease-of-use of the optimizer”. I disagree with both points. Regarding the performance aspect, I mentioned in my previous reply that I could come up with a method that takes the same amount of compute, relatively simple, but produces better results by using separate search spaces. Regarding “ease-of-use”, I don’t see how the current protocol reveals anything about the ease of use of an optimizer. Using the example that the authors brought up, how would we know of the existence of optimizer A or B if we don’t try other ranges? It seems like the only way to show such a property (which I agree would be very useful) is to refine the search spaces and observe that a consistent search space was chosen for one optimizer. Since we’re in the topic of examples, I can come up with another example to defend the refinement procedure: it could be the case that one optimizer outperforms all other optimizers, but only with parameter values that are outside the range of that chosen, but still something reasonable, so that a simple two-stage refinement would have revealed it. I think finding (or attempting to find) such examples (or the lack of) is the kind of contribution that I would be happy to accept as a paper.
> > > >
> > > > Lastly, I want to mention that I’m mainly criticizing the fixed search space approach, and not advocating for a specific tuning procedure. I brought up the 25-25 split two-stage procedure, because it seemed simple, and similar to what previous researches have done in the past. It could be 10-40, 20-30, or whatever that seems justifiable, as long as it is consistently done over all methods. The authors seem to justify not doing a refinement procedure by saying that it “require(s) additional human decisions and can thus introduce bias”, and it is a “debated” approach. Bias and debated-ness are all imprecise terms; we can’t possibly measure and compare the amount of bias introduced by two different methods. Therefore, I can only conclude that the appeal to using a fixed search space is to cater to a specific group of people, and also to make it “easier” to compare. I don’t think this is enough of a justification to not do search space refinements,  which should yield better performance in general, and could help answer questions like which optimizer is more “easy to tune”.
> > > >
> > > > All in all, I never expected a paper to satisfy everyone. But I also cannot accept a paper with a narrow scope, which could have easily been avoided. I would be happy to accept a future version of the paper that addresses the comments I made in my original review.

---

### Official Review · AnonReviewer1 · 2020-10-28

**Rating:** 4
**Confidence:** 5

**Review:**

### Summary
The authors of this paper conducted a thorough evaluation of deep learning optimizers across different compute budgets and learning rate schedules. They provide detailed analysis of the results. The design decisions are well-reasoned and explained throughout the paper.

### Comments
* As the authors note, there is certainly value in understanding the practical tradeoffs between optimizers: "for most algorithms, the only formal empirical evaluation is offered by the original work introducing the method"
* The writing is clear and easy to follow.
* Many of the findings are useful in the context of the DeepOBS dataset. For instance, Figure 3 highlights the diminishing returns of increasing the budget when tuning hyperparameters.
* Open-sourcing the data is great and beneficial for the community.

* The benchmark would benefit from a larger scale dataset(s). I'm not in favor of solely adopting DeepOBS, as past papers have shown systematic differences in evaluation at different scales [1][2]. Investigating whether there are systematic differences in optimization on larger problems e.g. machine translation or ImageNet would be valuable.
* As Reviewer 4 mentions, the momentum parameter should be tuned as 1 - \rho.

### Recommendation / Justification
I vote that this paper is below the acceptance threshold. There are many things to like about the approach taken is this paper, as highlighted above. However, the lack of larger scales datasets lessens the significance of the conclusions.

I'd increase my score if concerns about the datasets used were addressed. I understand it is challenging to do so during the rebuttal period, but I strongly believe that larger scale datasets would strengthen the work significantly.

### Minor feedback
* A tabular form of Figure 4 would improve clarity.
* I think it is worth acknowledging techniques for averaging the weights of neural networks, as these can have a substantial impact on final performance (Polyak averaging, exponential moving average, Stochastic Weight Averaging).
* I believe it is also worthwhile to benchmark a second-order optimizer. While the compute per step is more expensive, the comparison could be made fair by using the same compute budget for each optimizer.
* I am surprised by the choice of \alpha when tuning the lookahead optimizer. My suspicion is that tuning the momentum and learning rate is more fruitful than trying low values of \alpha.


[1] Frankle, Jonathan, Gintare Karolina Dziugaite, Daniel M. Roy, and Michael Carbin. "Stabilizing the lottery ticket hypothesis." arXiv preprint arXiv:1903.01611 (2019).

[2] Gale, Trevor, Erich Elsen, and Sara Hooker. "The state of sparsity in deep neural networks." arXiv preprint arXiv:1902.09574 (2019).


Edit: After the rebuttal period, I maintain my original rating but am increasing the confidence of my evaluation from 4 to 5. I thank the authors for their hard work and engaging in discussion. I agree with Reviewer 3 that tuning with a fixed seed and the lack of search space refinement is a major weakness. The lack of a larger dataset further limit the applicability of the results. As such, I do not believe the paper in its current form should be accepted to ICLR.

---

> ### Author Response · Authors · 2020-11-12
> **Response to the Comments of Reviewer 1**
>
> Dear Reviewer 1,
>
> thank you for providing a detailed review of our paper. We greatly appreciate that you called our work "a thorough evaluation of deep learning optimizers".
>
> We appreciate your comment about larger data sets. The size of the training problems, however, is another aspect where a benchmark has to find compromises. Training on ImageNet is significantly more costly in time and resources than on the architectures we used. So, given finite resources, opting for these large problems would have required us to reduce statistical fidelity or the number of optimizers to compare. In our opinion, our setup strikes a better balance. Of course one can always argue that one dimension or another in this balance should have been weighted differently, but we argue that such personal desiderata should not preclude the publication of our work. We agree that evaluating large-scale problems is an interesting avenue for further research in this area, similar to benchmarking optimizers on GANs or RL. We are happy to include a statement like this into our Limitation section, acknowledging that our benchmark is more applicable to small and medium-scale problems.
> It is also debatable whether large-scale problems are actually “typical” for the bulk of practitioners in real-world settings. Outside of computer vision, medium-size and even smallish datasets are not uncommon.
>
> We will address your minor feedback as much as possible. Adding a second-order optimizer would be interesting for future work. As you mentioned, it would require keeping the runtime of runs of different optimizer constant. This is tricky, even if exclusively using identical hardware, as the runtime can be affected by many factors. Providing the data of Figure 4 as a table or acknowledging techniques for averaging weights is certainly possible.
>
> We want to also invite you to engage in the discussion we have below with Reviewer 4, in which we are trying to find a constructive way to address as many requests made by you and the other reviewers as possible within the rebuttal phase.

---

> > ### Comment · AnonReviewer1 · 2020-11-21
> > **Response Thoughts**
> >
> > Thank you for your response. I have read through the other reviews and responses.
> >
> > I generally maintain my original thoughts about large datasets and agree with the comments made by reviewer 4 (I am also sympathetic of compute/time constraints). I think that omitting GANs and RL is fine as we'd expect different patterns of optimization in those domains. Including larger datasets is important for novelty/significance. I looked through Sivaprasad et al. (2020) again, and it seems like the current set of experiments extend their work primarily via the inclusion of more optimizers and learning rate schedules. They do draw the conclusion that Adam is the most practical optimizer in terms of ease to tune/performance, which is different from the conclusion in this paper.
> >
> > Overall, I think the changes and experiments you are conducting now definitely do strengthen the paper and make it more valuable to the community.

---

### Official Review · AnonReviewer2 · 2020-10-28
**Important, well executed, experiments with unfortunately no clear-cut outcome**

**Rating:** 6
**Confidence:** 4

**Review:**

This paper presents an extensive independent benchmark of 14 popular optimizers on a variety of deep learning tasks from DeepOBS (Schneider et al. 2019). They compare them at three different tuning budgets and with 4 learning rate schedules. The authors are realistic about their setup. They acknowledge that different people might have different desires for such a benchmark, and they are clear about the choices they made to keep the experiments feasible. While there is no clear-cut answer that tells practitioners which optimizer to use in what scenario and how to tune it, these experiments are valuable and I believe it is important that these results are shared with the community.

The quality of the presentation and the writing is good.

In terms of novelty, the authors model the target audience slightly differently from previous work (Schneider et al. 2019, Choi et al. 2019, Sivaprasad et al. 2020). I am not convinced that this approach is better per se than others, but a different angle and a different set of optimizers is a valuable contribution to the community. I believe that the description of (Sivaprasad et al. 2020) in Section 1.1. is not entirely accurate. They do not compare hyperparameter tuning methods, but rather benchmark optimizers similarly to this work at a continuum of hyperparameter tuning budgets (all with random search).

Finally, let me share two concerns:

1. The intro mentions three contributions: (i) performance varies greatly, (ii) trying different optimizers works as well as tuning a single one, (iii) they identify a significantly reduced subset of algorithms and parameter choices that perform well across experiments. Points (ii) and (iii) are interesting, but (ii) is formulated quite imprecisely and it is hard to see on which results this is based. I inspected Figures 9 to 12 in the Appendix and conclude "this might be true, but it is hard to see". I believe a quantative statement would be more useful/meaningful. Similarly, for point (iii) it is not clear from which results this is concluded, and what the high-performing subset is. Such a list would be valuable to many practitioners and should be clearly stated in the main text.

2. Figure 3 shows relative improvement across tasks. Any such measurements of 'improvement' are dependent on re-shifting or re-scaling of the loss, and are not necessarily meaningful when aggregated into a plot like this. Consider accuracy: measuring relative improvement (accuracy 1 / accuracy 2) would yield drastically different numbers than (error 1 / error 2 = (1-acc1) / (1-acc2)).

---

> ### Author Response · Authors · 2020-11-12
> **Response to the Comments of Reviewer 2**
>
> Dear Reviewer 2,
> thank you for your positive review of our work. We are happy that you agree with us that the paper offers important results to be shared with the community.
>
> We agree with your description of the work by Sivaprasad et al. (2020). We did not mean to characterize their work as a comparison of hyperparameter tuning methods, but we realize that our phrasing can be understood like this. We will re-formulate this part.
>
> Contribution (ii) is indeed based on Figure 2 and Figures 9 to 12 in the Appendix. Looking at a tuned optimizer, i.e. a single column, there is almost always a red (or at least white) cell in this column, indicating that there is a better-performing optimizer with default parameters.
> Whether this is a preferable strategy, however, depends on multiple factors, such as what the specific practitioner would consider "similar performance" or how much performance they are willing to trade-off for cheaper computation.
> In Section 3, we describe that Adam (and its variants) as well as AdaBound (and AMSBound) are optimizers that work well without tuning. Taking the better of them can often provide competitive results, even compared to tuned optimizers. If additional budget is available, adding a tuned version of Adam (or its variants) seems to be a good strategy. These conclusions are based mainly on Figure 2 and Figure 4, as well as the corresponding Figures in the appendix.
>
> Figure 3 is indeed not invariant to re-shifting and re-scaling. As such, it is best not read quantitatively, but qualitatively. What this figure shows is that tuning helps, but with diminishing returns, and also that there is a lot of underlying noise. Both statements also hold after re-scaling the used metric.
>
> We want to also invite you to engage in the discussion we have below with Reviewer 4, in which we are trying to find a constructive way to address as many requests made by you and the other reviewers as possible within the rebuttal phase.

---

### Author Response · Authors · 2020-11-21
**Updated Version**

Dear Reviewers,

we want to thank you again for your constructive feedback. We have just submitted an improved version of our paper, incorporating your suggestions.

* Most importantly, we re-ran a significant part of our benchmark, using the suggested momentum search space for all schedules and both tuning budgets (this is more than we initially thought we would be able to do in the rebuttal timeframe). Although this changed a few details, the overall statements of the paper remain unchanged.
* We have updated all results and plots in the paper as well as in the anonymous repository.
* We have also addressed Stochastic Weight Averaging, L2-regularization, and larger data sets in an added paragraph of the Limitations [SECTION 4], along with the more minor feedback such as our description of the work by Sivaprasad et al. [SECTION 1.1], the use of op. cit [SECTION 2.3], No Free Lunch Theorem [SECTION 1], etc.
* We added the extensive list of optimizers to our repository Readme with links to the respective papers and provided an additional tabular version of Figure 4 in Appendix H.

To address your requests we committed significant resources over the past days. You all seem to agree that this is important, valuable, and (most importantly) beneficial work to be shared with the community. A paper like this inevitably requires choices that can be criticized one way or another. We hope, though, that our efforts to address your requests have alleviated your concerns. Thanks again for your time and your comments!

---

> ### Comment · AnonReviewer4 · 2020-11-21
> **Updated score**
>
> Thank you very much for the heroic effort! I have updated my score accordingly.
>
> I believe that this work is exemplary in terms of being careful about baseline construction, something that is unfortunately too often overlooked in our field. Additionally, it rigorously highlights another important point that I believe many often overlook, that "there are now enough optimizers"; community effort should be diverted from introducing small variations around Adam and instead invest focus on more meaningful improvements in scaling machine learning optimization.

---

> > ### Comment · AnonReviewer2 · 2020-11-24
> > **Neutral sentiment**
> >
> > I have read the other reviews and the author's responses and maintain my (weak) recommendation for acceptance.
> >
> > In my opinion, this paper does not set a gold standard for optimizer benchmarking. The experimental design decisions and baselines are reasonable in some situations, yet inapplicable in others.
> > I do, however, have the same criticism towards previously published work in this area. Given the importance of good optimizer benchmarking and the long way we have to go, I think that the contribution of this work is valuable to the community and it should be shared.

---

### Decision · Program_Chairs · 2021-01-07
**Final Decision**

**Decision:**

Reject

**Comment:**

Contributions of this type are very important for the community. There is a great deal of confusion among practitioners about how to pick optimizers. Perhaps worse, there is confusion among optimization researchers about how to demonstrate the effectiveness of their novel algorithms on deep learning tasks. I applaud this paper as one of the best attempts to make sense of this confusion.

Unfortunately, I am recommending that it is rejected. This was an extremely difficult decision. This paper was very thoroughly discussed by reviewers, both with the authors and after the feedback phase. I agree with R4 that this paper is exemplary in terms of its breadth of optimizer choices. I also agree with R3 that this paper's choices regarding hyperparameter search spaces and seed fixing significantly diminish the contribution of the paper at hand. The key issue that persuaded my decision centered on whether the paper's evidence supported its conclusions.

The two key conclusions that I want to highlight are:

1. *evaluating multiple optimizers with default parameters works approximately as well as tuning the hyperparameters of a single, fixed optimizer*

2. *different optimizers exhibit a surprisingly similar performance distribution compared to a single method that is re-tuned or simply re-run with different random seeds*

These conclusions can only be supported if optimizers are well-tuned. Based on R3's remarks and a quick reading of the paper, I am concerned that the use of fixed search spaces means that these optimizers cannot be considered well-tuned. This concern splits into two sub-concerns.

1. I appreciate the author's desire to encode "no prior knowledge about well-working hyperparameter values". Unfortunately, I don't think this is realistic or possible. The learning rate range used in this paper did not include 1e100 for good reasons, all of which depend on the prior knowledge of our community. This isn't just a glib concern, the apparently neutral search spaces may bias the conclusions towards well-known methods whose hyperparameters are well-understood.

2. I am also skeptical of the choice to use the same range for hyperparameters with "similar naming". The reason is that these hyperparameters *may have been misnamed by the inventors* and may, in fact, play very different roles in the dynamics of optimization.

Top-line conclusions have a way of becoming memes in our community. Therefore, it is critical that conclusions, as stated, are actually supported by the experimental design and the empirical evidence. Unfortunately, I am not confident that this is is the case for the paper at hand.

It is clear that this paper represents a heroic effort by the authors. I am aware of the challenges involved in getting this type of paper published and of the urgent need for them. I hope that the authors address the concerns that I expressed and the concerns of the reviewers in a future submission.